# Gradient Descent Converges Linearly to Flatter Minima than Gradient Flow in Shallow Linear Networks

## Abstract

We study the gradient descent (GD) dynamics of a depth-2 linear neural network with a single input and output. We show that GD converges at an explicit linear rate to a global minimum of the training loss, even with a large stepsize–about 2/sharpness. It still converges for even larger stepsizes, but may do so very slowly. We also characterize the solution to which GD converges, which has lower norm and sharpness than the gradient flow solution. Our analysis reveals a trade off between the speed of convergence and the magnitude of implicit regularization. This sheds light on the benefits of training at the "Edge of Stability", which induces additional regularization by delaying convergence and may have implications for training more complex models.

## 1 Introduction

Training modern machine learning (ML) models like deep neural networks via empirical risk minimization (ERM) requires solving difficult high-dimensional, non-convex, under-determined optimization problems. Although they are usually intractable to solve in theory, we train models effectively in practice using algorithms like stochastic gradient descent (SGD). This highlights a disconnect between the worst-case convergence rate of SGD and its convergence on specific ERM problems that arise from training, e.g., neural networks. Even if we can solve the ERM problem, typical minimizers of the under-determined objective will overfit and generalize poorly. That said, the specific solutions found by SGD and its variants usually *do* successfully generalize. Understanding how and why we are able to successfully optimize and generalize with these models is of great interest to the ML community and could help fuel continued progress in applied ML.

A key feature of popular ML models, including neural networks, is that the model output is related to the product of model parameters in successive layers. For instance, the output of a 2 layer feedforward network with ReLU activations has output $\mathbf{W}_2 \operatorname{ReLU}(\mathbf{W}_1 x + \mathbf{b}_1)$, which is closely related to the product of the weight matrices $\mathbf{W}_2\mathbf{W}_1$. Ultimately, this "self-multiplication" of different model parameters gives rise to the non-convex and under-determined ERM problems that cause such (theoretical) difficulties.

In this work, we distill this parameter self-multiplication property down to its simplest form and comprehensively explain how it affects the training optimization dynamics, the "implicit regularization" of the model parameters, and the "edge-of-stability" dynamics that arise in certain regimes. In particular, we consider the extremely simple problem of learning a univariate linear model $\hat{y} = mx$ to minimize the squared error, except we parameterize the slope as $m = m(\mathbf{a}, \mathbf{b}) = \mathbf{a}^\top \mathbf{b}$ in terms of self-multiplying parameters $\mathbf{a}, \mathbf{b} \in \mathbb{R}^d$. This can also be thought of as a depth-2 linear neural network with $d$ hidden units. For training data $\{(x_i, y_i) \in \mathbb{R} \times \mathbb{R}\}_{i=1}^n$, this results in the loss

$$\min_{\mathbf{a},\mathbf{b}\in\mathbb{R}^d} \bar{L}(\mathbf{a}, \mathbf{b}) \quad := \quad \frac{1}{2n} \sum_{i=1}^n \left(\mathbf{a}^\top \mathbf{b} x_i - y_i\right)^2. \tag{1}$$

This objective is equivalent—by rescaling and subtracting a constant—to the even simpler loss[1]

$$\min_{\mathbf{a},\mathbf{b}\in\mathbb{R}^d} L(\mathbf{a},\mathbf{b}) \quad := \quad \frac{1}{2}(\mathbf{a}^\top \mathbf{b} - \Phi)^2. \tag{2}$$

In what follows, we focus on this formulation and assume $\Phi \geq 0$ w.l.o.g. for simplicity and clarity.

Despite its simplicity, the objective (2), which has also been studied by prior work (Lewkowycz et al., 2020; Wang et al., 2022; Chen & Bruna, 2023; Ahn et al., 2024; Xu & Ziyin, 2024), is a useful object of study because it has a number of qualitative similarities to more complex and realistic problems like deep learning training objectives. First, it has similar high-level properties—the problem (2) is non-convex and highly under-determined because the set of minimizers constitutes the $(2d - 1)$-dimensional hyperboloid in $\mathbb{R}^{2d}$ that solves $\mathbf{a}^\top \mathbf{b} = \Phi$. It also exhibits some of the same symmetries as realistic neural networks; for example, $\mathbf{a}^\top \mathbf{b}$ is invariant to swapping "neurons" $(\mathbf{a}_i, \mathbf{b}_i) \leftrightarrow (\mathbf{a}_j, \mathbf{b}_j)$ or to rescaling $(\mathbf{a}_i, \mathbf{b}_i) \rightarrow (c\mathbf{a}_i, c^{-1}\mathbf{b}_i)$. More importantly, the dynamics when optimizing (2) with gradient descent are qualitatively similar to the dynamics of training more complex models (see, e.g., Xu & Ziyin, 2024). Simultaneously, the problem (2) is simple enough that we can provide a detailed and nearly comprehensive characterization of several different aspects of training. Specifically, we will present the following results:

**Convergence of gradient descent.**   Despite the non-convexity of the optimization problem (2), prior work has shown that GD converges to a global minimum from a.e. initialization (Wang et al., 2022). We show that, in fact, it converges at a linear rate. In addition, we identify several phases that depend on the relationship between the stepsize, $\eta$; the scale of the parameters, $\boldsymbol{\lambda} := \|\mathbf{a}\|^2 + \|\mathbf{b}\|^2$; and the residuals, $\boldsymbol{\varepsilon} := \mathbf{a}^\top \mathbf{b} - \Phi$. Several of these phases are closely related to the so-called Edge of Stability (EoS) phenomenon (Cohen et al., 2021), where gradient descent decreases the objective (although non-monotonically) despite the largest eigenvalue of the objective's Hessian matrix being larger than the critical threshold $2/\eta$.

**Location of convergence.**   In addition to showing how fast gradient descent converges to *some* global minimizer, we can also identify key properties of the model related to *which* of the many possible solutions, $\mathbf{a}^\top \mathbf{b} = \Phi$, gradient descent will converge to. To do so, we show that gradient descent implicitly regularizes the "imbalance" of the parameter vectors, quantified by $Q := \sum_{i=1}^d \left|\mathbf{a}_i^2 - \mathbf{b}_i^2\right|$, with a larger stepsize generally leading to stronger regularization. This is notably different from the behavior of Gradient Flow (GF) (the $\eta \rightarrow 0$ limit of gradient descent), which *conserves* $Q$. Since GF is often employed in the literature as an easier-to-analyze approximation of gradient descent (e.g. Du et al., 2018; Tarmoun et al., 2021), our results highlight a potential danger of over-reliance on this approximation. The balance of the parameters is also closely related to the "sharpness" of the solution, i.e. the maximum eigenvalue of the Hessian, which is equal to $\boldsymbol{\lambda} := \|\mathbf{a}\|^2 + \|\mathbf{b}\|^2$ at solutions $\mathbf{a}^\top \mathbf{b} = \Phi$. For the problem (2) specifically, the actual prediction function defined by any solution $\mathbf{a}^\top \mathbf{b} = \Phi$ is the same—after all at any minimizer, $\hat{y} = \Phi x$ regardless of the parameters—so the sharpness is not relevant to generalization. Nevertheless, there is a large body of work in other contexts showing that less sharp minima of the loss tend to generalize better (Hochreiter & Schmidhuber, 1997; Keskar et al., 2016; Smith & Le, 2018; Park et al., 2019), and our analysis shows how the self-multiplying structure of (2) tends to regularize the sharpness.

The key to our analysis is the following pair of observations. On the one hand, gradient descent iterations change the imbalance like $Q(t + 1) = |1 - \eta^2 \boldsymbol{\varepsilon}(t)^2| Q(t)$, so the imbalance decreases throughout optimization for $0 < \eta < \sqrt{2}/|\boldsymbol{\varepsilon}(t)|$. At the same time, the objective $L$ does *not* globally satisfy the Polyak-Łojasiewicz (PL) condition (Polyak, 1963) because the origin is a saddle point, but it *does* satisfy a version of the PL condition along the GD trajectory (see Definition 2), which is sufficient to prove linear convergence of GD to a global minimizer. Interestingly, the PL constant along the GD trajectory, which controls the speed of convergence, is equal to the smallest value of $\boldsymbol{\lambda}(t)$ encountered along the way, which is itself approximately equal to the value of $Q(t)$ at the first time that $\mathbf{a}(t)^\top \mathbf{b}(t) > 0$. Thus, the *stronger* the implicit regularization of $Q$, the *slower* the convergence of GD, and vice versa, which puts these goals directly at odds with each other.

---

[1]See Lemma 4 in Appendix A for a simple proof.

RELATED WORK

A large body of research has shown empirically that training neural networks with larger learning rates tends to lead to better generalization (LeCun et al., 2002; Bjorck et al., 2018; Li et al., 2019; Jastrzebski et al., 2020). However, in classical settings, convergence can only be guaranteed when the stepsize is small enough that $\lambda_{\max}(\nabla^2 L) < 2/\eta$ throughout optimization (Bottou et al., 2018). Nevertheless, a recent line of work starting with Cohen et al. (2021) observed that when training neural networks, the maximum eigenvalue of the Hessian, or "sharpness", tends to grow throughout training until it reaches, or even surpasses the critical $2/\eta$ threshold. But rather that diverging, the loss continues to decrease (non-monotonically) while the sharpness continues to hover around $2/\eta$, which is referred to as the Edge of Stability (EoS) phenomenon. Understanding more deeply the training of neural networks with large stepsizes is of great interest.

Problems closely resembling (2) have been studied previously. Ahn et al. (2024) study losses of the form $\ell(ab)$ with $a, b \in \mathbb{R}$ and $\ell$ any convex, Lipschitz, and even function. The assumption that $\ell$ is even means it is minimized at zero (this is analogous to $\Phi = 0$ in our case), and they prove convergence to zero from any initialization with any stepsize, but without a rate. However, this result relies crucially on both the loss being Lipschitz and minimized at zero—and this is not surprising, we know that GD diverges on realistic objectives when the stepsize is too large. They also show that the limit point of gradient descent satisfies $|a_\infty^2 - b_\infty^2| \approx \min\{2/\eta, |a_0^2 - b_0^2|\}$, i.e. the imbalance between the weights is implicitly regularized down to the level of $2/\eta$. Chen & Bruna (2023) study (2) with scalar $a, b \in \mathbb{R}$ and prove that the limit point of GD satisfies $a - b \to 0$ when the stepsize is chosen slightly too large for convergence to any minimizer $ab = \Phi$ to be possible. This is qualitatively similar to our work, but they intentionally choose a too-large stepsize in order to highlight the implicit regularization of the imbalance, while we provide conditions on $\eta$ under which convergence to a minimizer and some amount of regularization happen simultaneously.

In a related study, Xu & Ziyin (2024) explore the continuous dynamics of gradient flow using the exact same model discussed here. They demonstrate that the dynamics unfold along a one-dimensional curve, with the location of convergence distinctly defined by conserved quantities. Contrary to their findings, our research reveals this is not the case for gradient descent, highlighting the danger of relying excessively on continuous models to understand discrete non-convex optimization dynamics.

In the most closely related work, Wang et al. (2022) study the exact objective (2) and show that gradient descent using any stepsize up to $\eta \lesssim 4/\text{sharpness}$—approximately twice as large as the classical threshold of $2/\text{sharpness}$—eventually converges to a minimizer, but without a rate. They also show some level of implicit regularization of $\lambda$, e.g. at convergence $\lambda \leq \frac{2}{\eta}$. In comparison, we provide an explicit convergence rate for GD and give a more detailed connection between this rate and the implicit regularization.

Finally, many papers have studied other models such as matrix factorization or linear neural networks (Saxe et al., 2014; Arora et al., 2019; Gidel et al., 2019; Tarmoun et al., 2021; Xu et al., 2023; Nguegnang et al., 2024), which are more faithful representations of realistic neural networks, but they are also much more difficult to analyze. Due to this difficulty, these results often only apply to gradient flow, or to GD with a very small learning rate, or to GD under additional, hard to interpret assumptions. In this work, we focus on the problem (2) in order to obtain a simpler, easier to interpret set of results.

## 2 NOTATIONS AND SETTING

The gradient descent dynamics of the parameters are described by

$$\begin{bmatrix} \mathbf{a}(t+1) \\ \mathbf{b}(t+1) \end{bmatrix} = \begin{bmatrix} \mathbf{a}(t) \\ \mathbf{b}(t) \end{bmatrix} - \eta \nabla L(\mathbf{a}(t), \mathbf{b}(t)) = \begin{bmatrix} \mathbf{a}(t) \\ \mathbf{b}(t) \end{bmatrix} - \eta(\mathbf{a}(t)^\top \mathbf{b}(t) - \Phi) \begin{bmatrix} \mathbf{b}(t) \\ \mathbf{a}(t) \end{bmatrix} \quad (3)$$

However, tracking the dynamics of $\mathbf{a}$ and $\mathbf{b}$ is somewhat unwieldy due to the overparametrization of the model $\mathbf{a}^\top \mathbf{b}$. Therefore, it will be convenient to reparametrize the dynamics in terms of the following three quantities, rather than the parameters themselves.

**Residuals.** We define the residuals, $\varepsilon := \mathbf{a}^\top \mathbf{b} - \Phi = \pm\sqrt{2L(\mathbf{a}, \mathbf{b})}$, which measure the distance to the manifold of minima. We will control the magnitude of $\varepsilon$ in order to prove convergence.

**Norm of the parameters.** We denote by $\boldsymbol{\lambda} := \|\mathbf{a}\|^2 + \|\mathbf{b}\|^2$ the squared Euclidean norm of the parameter vectors. This quantity is relevant because the Hessian of the loss at a solution $\mathbf{a}^\top \mathbf{b} = \Phi$ is

$$\nabla^2 L(\mathbf{a}, \mathbf{b}) \;=\; \begin{bmatrix} \mathbf{b} \\ \mathbf{a} \end{bmatrix} \begin{bmatrix} \mathbf{b} \\ \mathbf{a} \end{bmatrix}^\top + \varepsilon \begin{bmatrix} 0 & I \\ I & 0 \end{bmatrix} = \begin{bmatrix} \mathbf{b} \\ \mathbf{a} \end{bmatrix} \begin{bmatrix} \mathbf{b} \\ \mathbf{a} \end{bmatrix}^\top \tag{4}$$

so it is rank-1 and has top eigenvalue, or "sharpness", equal to $\boldsymbol{\lambda}$. Also, by the Cauchy-Schwarz and Young inequalities, at any solution $\Phi = \mathbf{a}^\top \mathbf{b} \leq \|\mathbf{a}\|\|\mathbf{b}\| \leq \frac{1}{2}(\|\mathbf{a}\|^2 + \|\mathbf{b}\|^2) = \frac{1}{2}\boldsymbol{\lambda}$, with equality occurring when $\mathbf{a} = \mathbf{b}$. Therefore, the minimum sharpness of any solution is $2\Phi$. Moreover, this shows that on this model the lowest norm and the flattest solutions coincide.

The scale $\boldsymbol{\lambda}$ is also useful in our analysis because the evolution of the residuals due to (3) is closely related to $\boldsymbol{\lambda}$:

$$\varepsilon(t+1) \;=\; \varepsilon(t)\big(1 - \eta\boldsymbol{\lambda}(t) + \eta^2\varepsilon(t)\big(\varepsilon(t) + \Phi\big)\big). \tag{5}$$

Thus, the term $-\eta\boldsymbol{\lambda}$ is the main cause of the decrease in the magnitude of $\varepsilon$. Finally, the GD dynamics of $\varepsilon$ and $\boldsymbol{\lambda}$ are completely determined by each other

$$\boldsymbol{\lambda}(t+1) \;=\; \big(1 + \eta^2\varepsilon(t)^2\big)\boldsymbol{\lambda}(t) - 4\eta\varepsilon(t)\big(\varepsilon(t) + \Phi\big). \tag{6}$$

**The imbalance.** To complement the above, we also define $Q_i := \mathbf{a}_i^2 - \mathbf{b}_i^2$ and $Q := \sum_{i=1}^d |Q_i|$. The gradient descent dynamics on the $Q_i$'s due to (3) is described by

$$Q_i(t+1) \;=\; \big(1 - \eta^2\varepsilon(t)^2\big) \cdot Q_i(t) \quad \Longrightarrow \quad Q(t+1) \;=\; \big|1 - \eta^2\varepsilon(t)^2\big| \cdot Q(t). \tag{7}$$

From the lack of a term linear in $\eta$, we can see that the $Q_i$'s are conserved by gradient flow, see Eq. 8, but *not* by gradient descent, which decreases their magnitude for any $\eta < \sqrt{2}/|\varepsilon|$. This is the essential cause of GD's implicit regularization effect. In our analysis, we use $Q$ in two ways: (i) the lower bound $\boldsymbol{\lambda} \geq Q$ is key our the proof of GD's convergence speed, and (ii) we use $Q$ to help characterize the location of convergence.

## 3 LOCATION OF CONVERGENCE

Our first result concerns which solution is reached by gradient descent:

**Theorem 1.** *For $\eta < \min\left\{\frac{1}{2|\varepsilon(0)|}, \frac{2}{\sqrt{\boldsymbol{\lambda}(0)^2 + 4\Phi^2}}\right\}$, at the limit point of gradient descent* [2]

$$0 \;<\; |Q_i(0)| \exp\left(-\frac{\sqrt{\eta}\varepsilon(0)^2}{\Phi}\right) \;<\; |Q_i(\infty)| \;<\; |Q_i(0)| \exp\left(-\eta^2 \sum_{t=0}^{\infty} \varepsilon(t)^2\right) \;<\; |Q_i(0)|.$$

Theorem 1 follows from the iteration of Eq. (7) to describe the evolution of $Q_i$ with gradient descent. The first bound on the step size is needed to prove the lower bound on $|Q_i(\infty)|$, and the second to show rapid convergence, as detailed in Theorem 2. A full proof is located in Appendix H. For a geometric intuition on why gradient descent reduces the imbalance compared to gradient flow, see Figure 1. Indeed, gradient flow conserves the quantities $Q_i$ by curving away from the origin, indeed:

$$\dot{Q}_i \;=\; 2(\mathbf{a}_i\dot{\mathbf{a}}_i - \mathbf{b}_i\dot{\mathbf{b}}_i) \;=\; 2(\mathbf{a}_i(-\varepsilon\mathbf{b}_i) - \mathbf{b}_i(-\varepsilon\mathbf{a}_i)) \;=\; 0. \tag{8}$$

The discretization error, introduced by the fact that gradient descent moves along the parallel vector to the curve, results in GD moving "inward" towards the line $\mathbf{a} = \mathbf{b}$, resulting in a smaller imbalance. Theorem 1 thus implies the following message:

**Takeaway 1:** *Gradient descent converges to a solution with lower imbalance than gradient flow, although the imbalance always remains non-zero.*

Theorem 1 unveils and describes an implicit regularization effect which is only due to the action of discretizing the dynamics. Given that GF is frequently used as a simpler analytical stand-in for gradient descent in the literature, Takeaway 1 underscores the risks associated with over-relying on this approximation, potentially leading to inaccurate predictions about real-world behaviors.

---

[2] A similar result holds for larger stepsizes, at the cost of a more complicated statement. See Appendix H.

Furthermore, Eq. (7) and Theorem 1 allow us to roughly quantify this implicit regularization effect when the product between the learning rate and the residuals is small:

$$Q(\infty) \quad = \quad Q(0) \prod_{t=0}^{\infty} \left|1 - \eta^2 \varepsilon(t)^2\right| \quad \approx \quad \exp\left(-\eta^2 \sum_{t=0}^{\infty} \varepsilon(t)^2\right). \tag{9}$$

This demonstrates that the degree of regularization on $Q$ is determined by the rate of loss reduction. If the loss decreases quicker, then $\sum_{t=0}^{\infty} \varepsilon(t)^2$ is smaller and $Q(\infty)$ is closer to $Q(0)$. Conversely, if the loss decreases at a slower pace then $\sum_{t=0}^{\infty} \varepsilon(t)^2$ is larger and $Q(\infty)$ is much smaller. In the next section, we will show that for the stepsize described in Theorem 1, the loss actually converges to zero at a linear rate, so $\sum_{t=0}^{\infty} \varepsilon(t)^2 \approx \varepsilon(0)^2/\eta\mu$ for an certain value of $\mu$.

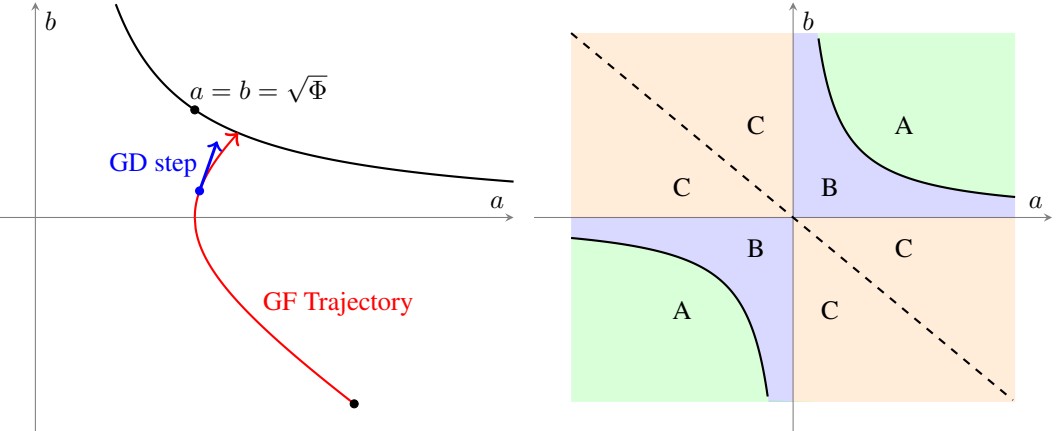

Figure 1: The GF trajectory curves away from the origin, so the discretization error of each GD step brings it closer to the line $a = b$.

Figure 2: The qualitative behavior of GD steps differs in each of the three regions.

## 4 SPEED OF CONVERGENCE

Our second result is a characterization of the convergence speed of gradient descent.

**Theorem 2.** *Let* $\eta < \min\left\{\frac{1}{2|\varepsilon(0)|}, \frac{2}{\sqrt{\boldsymbol{\lambda}(0)^2+4\Phi^2}}\right\}$, *denote* $\bar{\eta} := \min\left\{\eta, \frac{2}{\sqrt{\boldsymbol{\lambda}(0)^2+4\Phi^2}} - \eta\right\}$. *Then if* $Q(0) \neq 0$, *for any* $\delta > 0$, *gradient descent reaches a point* $L(\mathbf{a}(T), \mathbf{b}(T)) \leq \delta$ *with*

$$T \leq \mathcal{O}\left(\frac{\max\{\log|\varepsilon(0)|, 0\}}{\bar{\eta}Q(0)\exp(\min\{-\mathbf{a}(0)^\top\mathbf{b}(0), 0\})} + \frac{\log\frac{1}{\delta}}{\bar{\eta}Q(0)\exp(\min\{-\mathbf{a}(0)^\top\mathbf{b}(0), 0\}) + \bar{\eta}\Phi}\right).$$

*If* $\min\left\{\frac{\sqrt{2}}{\varepsilon}, \frac{2}{\sqrt{\boldsymbol{\lambda}(0)^2+4\Phi^2}}\right\} < \eta < \min\left\{\frac{2}{\varepsilon}, \frac{2}{\boldsymbol{\lambda}} + \frac{2\varepsilon(\varepsilon+\Phi)}{\boldsymbol{\lambda}^3}\right\}$ *we have convergence but it could be logarithmically slow.*

A full proof can be found in Appendix F. The key idea of the proof is to show from the update equation of the residuals, (5), that roughly speaking

$$\varepsilon(t+1) \approx (1 - \eta\boldsymbol{\lambda}(t))\varepsilon(t). \tag{10}$$

Since we also show that $0 < \eta\boldsymbol{\lambda}(t) < 2$ for all $t$, this means that GD will converge at a linear rate $(1 - \eta\min_t \boldsymbol{\lambda}(t))$. However, care must be taken to lower bound the smallest parameter norm $\boldsymbol{\lambda}(t)$ visited by GD because if the iterates stray too close to the origin, where $\boldsymbol{\lambda} = 0$, then this linear convergence could be arbitrarily slow. Therefore, the main technical challenge is to bound the GD iterates away from the origin. The key observation is that $\boldsymbol{\lambda} \geq Q$, and we can easily track the evolution of $Q$ using (7) (the updates on $\boldsymbol{\lambda}$, (6), are more difficult to control).

To lower bound $\boldsymbol{\lambda}$ along the trajectory, we break the parameter space into three regions, as depicted in Figure 2. In region A, by the Cauchy-Schwarz and Young inequalities, $2\Phi < 2\mathbf{a}^\top\mathbf{b} \leq 2\|\mathbf{a}\|\|\mathbf{b}\| \leq$

$\|\mathbf{a}\|^2 + \|\mathbf{b}\|^2 = \boldsymbol{\lambda}$. So, within region A we have linear convergence with rate $(1 - 2\eta\Phi)$. Region B contains points where $\boldsymbol{\lambda}$ is arbitrarily small, but since $\boldsymbol{\varepsilon} < 0$ but $\mathbf{a}^\top \mathbf{b} > 0$ here, by (6)

$$\boldsymbol{\lambda}(t+1) = \left(1 + \eta^2 \boldsymbol{\varepsilon}(t)^2\right)\boldsymbol{\lambda}(t) - 4\eta \underbrace{\boldsymbol{\varepsilon}(t)}_{<0} \underbrace{\mathbf{a}(t)^\top \mathbf{b}(t)}_{>0} \geq \left(1 + \eta^2 \boldsymbol{\varepsilon}(t)^2\right)\boldsymbol{\lambda}(t). \tag{11}$$

Thus, within region B, GD *increases* the value of $\boldsymbol{\lambda}$, so we can lower bound $\boldsymbol{\lambda}(t) \geq \boldsymbol{\lambda}(\tau)$, where $\tau$ is the time that the GD iterates entered region B. Furthermore, we always have $\boldsymbol{\lambda}(\tau) \geq Q(\tau)$, so all that is needed is to control $Q(\tau)$ at the time that GD enters region B. Finally, to address region C, the $Q$ updates (7) show $Q(t+1) = (1 - \eta^2 \boldsymbol{\varepsilon}(t)^2)Q(t)$ while we prove that $\boldsymbol{\varepsilon}(t+1) \approx (1 - \eta Q(t))\boldsymbol{\varepsilon}(t)$. We use this to argue that the time it takes to substantially decrease $Q$ scales with $\eta^{-2}$ while the time to exit region C by making $\boldsymbol{\varepsilon}(t+1) > -\Phi$ scales with only $\eta^{-1}$. So at the time that GD exits region C, $Q(\tau)$ remains only slightly smaller than $Q(0)$.

Putting this all together, if GD is initialized in region C, it takes $O(\eta^{-1})$ steps to leave, at which point it enters region B with $Q(\tau) \approx Q(0)$, which serves as a lower bound on $\boldsymbol{\lambda}(t)$ until convergence, assuming that the remaining iterates stay in region B. If GD is initialized and remains in region A, then $\boldsymbol{\lambda}(t) \geq 2\Phi$ throughout optimization. To complete the argument, we account for the case where the GD trajectory leaves and/or re-enters a region more than once.

The structure of the proof—which relies on lower bounding the imbalance, $Q(\tau)$, of the GD iterate closest to the origin—is essentially a mirror image of the proof of Theorem 1, and leads to another key takeaway:

**Takeaway 2:** *The stronger the implicit regularization of the imbalance, Q, the slower the convergence and vice versa.*

The trade-off between the convergence speed and implicit regularization of $Q$ is illustrated by Figure 3. This experiment also indicates that in the EoS regime where the stepsize is larger than what is allowed by Theorem 2 but smaller than approximately $4/\boldsymbol{\lambda}(0)$, GD still converges to a solution, but both the rate of convergence and the amount of regularization of $Q$ have a intricate and chaotic dependence on the initialization and stepsize. Nonetheless, also in this EoS regime we see that that the convergence rate and amount of regularization have a generally negative relationship.

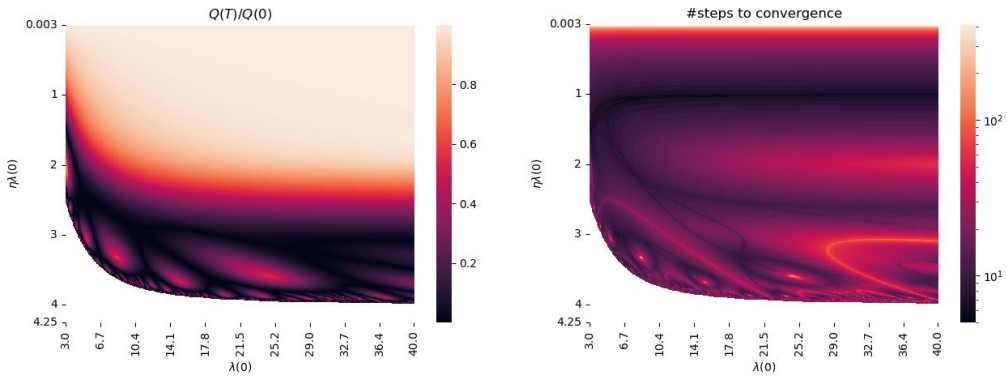

Figure 3: We used GD to minimize the problem (2) with $\Phi = 1$ using different stepsizes from various initializations with $\boldsymbol{\varepsilon}(0) = -2$ held constant. The x-axis corresponds to the initial scale $\boldsymbol{\lambda}(0)$ of the initialization, while the y-axis corresponds to the $\eta\boldsymbol{\lambda}(0)$. For stepsizes $\eta \lesssim 2/\boldsymbol{\lambda}(0)$, the amount of regularization $Q(T)/Q(0)$ is limited, but convergence is quick, and quickest around $\eta \approx 1/\boldsymbol{\lambda}(0)$. On the other hand, for very large stepsizes $\eta > 2/\boldsymbol{\lambda}(0)$, convergence is more chaotic; the convergence rate and regularization have a negative relationship but depend on $\eta$ and $\boldsymbol{\lambda}(0)$.

## 5    ON THE STEP SIZE AND THE EDGE OF STABILITY

We characterize here the regime of the dynamics of $\boldsymbol{\varepsilon}$ given the size of the learning rate and we sketch a proof for that. We conclude that for some learning rate higher than the bound of Theorem

2 convergence still happens, although logarithmically fast. This may shed light on what happens at the edge of stability: Convergence happens but slow, the imbalance gets zeroed out.

## 5.1 THE STABLE REGIME

We know from Theorem 2 that $\eta < 2/\boldsymbol{\lambda}$ along the all trajectory implies linear convergence and we manage to bound $\boldsymbol{\lambda}$ along the whole trajectory thanks to the following lemma and a characterization of the effect of discretization. These two lemmas are proved in Appendix C[3].

**Lemma 1.** *Let $\eta \leq 1/\boldsymbol{\varepsilon}(0)$ and assume that $|\boldsymbol{\varepsilon}(t)|$ is monotonically decreasing along the trajectory of GD. Then $\boldsymbol{\lambda}(t)$ is bounded for all steps $t$ by*

$$\boldsymbol{\lambda} \quad \leq \quad \sqrt{\boldsymbol{\lambda}(0)^2 + 4\Phi^2}.$$

*Analogously, along the trajectory of GF, $\boldsymbol{\lambda}(t)$ is bounded for all steps $t$ by the same quantity.*

Instrumental to prove this theorem is the observation that:

**Lemma 2.** *The quantity $\boldsymbol{\lambda}^2 - 8\varepsilon(\varepsilon + \Phi) + 4\varepsilon^2$ is conserved by the gradient flow on $L$, and it is reduced by gradient descent as long as $\eta \leq \min\{1/\boldsymbol{\varepsilon}(0), 2/\boldsymbol{\lambda}\}$.*

## 5.2 THE CASE OF EDGE OF STABILITY

Recent findings by Cohen et al. (2021) demonstrate that during the training of neural networks with full batch gradient descent at a step size of $\eta$, the largest eigenvalue of the Hessian stabilizes right above $2/\eta$. This is somewhat surprising as when the gradients are a linear function of the parameters, if $\eta > \frac{2}{\boldsymbol{\lambda}}$, gradient descent diverges. A very good exemplification of this fact is the case of one dimensional parabola, see Cohen et al. (2021). However, neural networks, surprisingly, convergence even if $\eta \geq \frac{2}{\boldsymbol{\lambda}}$. Our results here show that the reason my be the product structure and its interaction with discrete dynamics. Indeed, our results show how convergence happen also for $\eta > 2/\boldsymbol{\lambda}$ but slower. This implies that training happens even though at the edge of stability. Moreover, we prove that the slower the training and the bigger the learning rate, the lower the parameter norm of the solution gradient descent will eventually converge to. This suggests that training at the edge of stability may induce increased implicit regularization.

In the case in which the learning rate is slightly bigger than threshold above $\eta > \frac{2}{\sqrt{\boldsymbol{\lambda}(0)^2+4\Phi^2}}$ we show that convergence still happen but at a lower speed. This can be noticed in the case of Figure 3. Precisely, define $\tilde{\eta}$ as

**Definition 1.** *Let $\mathbf{a}, \mathbf{b} \in \mathbb{R}^n$, denote by $\tilde{\eta}(\mathbf{a}, \mathbf{b})$ the value*

$$\tilde{\eta}(\mathbf{a}, \mathbf{b}) := \frac{2}{\boldsymbol{\lambda}} \left(1 + \alpha + 4\alpha^2 + 20\alpha^3 + 112\alpha^4 + O(\alpha^5)\right) \qquad \text{with} \quad \alpha := \frac{\varepsilon(\varepsilon + \Phi)}{\boldsymbol{\lambda}^2}.$$

Then we have that

**Proposition 1.** *Let $\frac{2}{\sqrt{\boldsymbol{\lambda}(0)^2+4\Phi^2}} \leq \eta < \tilde{\eta}$ gradient descent on $L$ converges. However, convergence may happen at only logarithmic speed.*

In the case in which $\eta$ is even bigger and approaches $4/\boldsymbol{\lambda}$, Theorem 1 of Wang et al. (2022) suggests that there exists a time in which $\eta \leq \frac{2}{\sqrt{\boldsymbol{\lambda}(0)^2+4\Phi^2}}$. Just applying our result then we ensure linear convergence, although from that time on. As we showed above, this initial oscillatory phase may, however, last for arbitrarily long time.

## 6 CONVERGENCE ANALYSIS

In this section we present the notion we introduced to prove Theorem 2 and connect the speed of convergence with a lower bound on $\boldsymbol{\lambda}$ along the trajectory.

---

[3]Lemma 1 is a direct consequence of Lemmas 7 and 9, Lemma 2 of Lemmas 6 and 8.

## 6.1 PL Condition Along the Trajectories

To assess convergence rates, we introduce a convergence criterion we call the Polyak-Łojasiewicz Condition Along the Trajectories (PLAT Condition). This criterion serves as a generalization of the traditional PL condition. Specifically adapted to non-convex settings where empirical data indicate quick convergence although the problem itself is not necessarily PL.

**Definition 2** (PL Condition Along the Trajectories (PLAT)). Consider the optimization problem $minimize_x f(x)$ for $f \colon S \subseteq \mathbb{R}^d \to \mathbb{R}$, paired with an optimization algorithm $\mathcal{A}$. This problem-algorithm pair satisfies the PLAT at $x_0 \in \mathbb{R}^d$ if there exists a constant $\mu(x_0) > 0$ and a stationary point $f^*$ such that for all points $x$ visited by $\mathcal{A}$, $\frac{1}{2}\|\nabla_x f(x)\| \geq \mu(x_0) \cdot \big(f(x) - f^*\big)$.

While a function $f$ satisfying the traditional PL condition implies it meets the PLAT criteria when equipped with gradient descent (GD) and gradient flow (GF) for every initialization. Notably, however, if a function is PLAT with GD and GF almost everywhere, it may admits saddle points. Almost everywhere linear convergence to global minima is anyway ensured.

## 6.2 Dynamics and Implications

The introduction of the PLAT criterion facilitates a nuanced understanding of optimization trajectories. In the context of gradient descent with step size $\eta$, satisfying PLAT at $x_0$ means that the function, when restricted to the trajectory of the algorithm,

$$\big\{x \text{ s.t. } \exists n \in \mathbb{N} \text{ s.t. } x = X_n \text{ where } X_0 = x_0 \text{ and } X_{k+1} := X_k - \eta \nabla f(X_k)\big\}$$

adheres to the PL condition: $\frac{1}{2}\|\nabla_x f(x)\| \geq \mu(x_0) \cdot \big(f(x) - f^*(x_0)\big)$.

This, thus, primarily concerns the speed of linear convergence which we lower bound by $\mu$, not whether convergence happen. Note also that the proof of the fact that PL-condition implies exponential convergence for gradient flow and linear convergence for gradient descent is actually subtly using the weaker assumption of PL-condition along trajectories, see, e.g., (Karimi et al., 2016, Theorem 1).

## 6.3 Warm Up: Gradient Flow on our Problem

We illustrate that the loss function $L(\mathbf{a}, \mathbf{b})$ we take into consideration when equipped with gradient flow (GF), satisfies PLAT almost everywhere. As a first step, note that for any $\mathbf{a}, \mathbf{b} \in \mathbb{R}^d$ we have

**Lemma 3** ($L$ is locally PL). *$L$ admits local PL constant $\mu(\mathbf{a}, \mathbf{b}) = \|a\|^2 + \|b\|^2 = \boldsymbol{\lambda}$.*

$$\frac{1}{2}\|\nabla L(\mathbf{a}, \mathbf{b})\|^2 = (\|\mathbf{a}\|^2 + \|\mathbf{b}\|^2) \cdot \frac{1}{2}\big(\mathbf{a}^\top \mathbf{b} - \Phi\big)^2 = \boldsymbol{\lambda} \cdot L(\mathbf{a}, \mathbf{b}).$$

However, $L$ does not meet the PL condition globally as $\boldsymbol{\lambda} = 0$ at the saddle point $\alpha, \beta = 0$.

Next note that gradient flow conserves the quantity $Q(\mathbf{a}, \mathbf{b}) := \sum_{i=1}^{n} |a_i^2 - b_i^2|$ which is always smaller than $\boldsymbol{\lambda}$, and the equality is reached only if for all $i \in \{1, 2, \ldots, n\}$ at least one between $\mathbf{a}_i$ and $\mathbf{b}_i$ is equal to zero, more formally $Q = \boldsymbol{\lambda}$ if and only if $\sum_i \min\{|\mathbf{a}_i|, |\mathbf{b}_i|\} = 0$. This implies that if at initialization $\mathbf{a}, \mathbf{b}$ satisfy $Q(\mathbf{a}, \mathbf{b}) > 0$, then $L$ and gradient flow are PLAT with constant $Q$. However, we can reach to similar conclusions also if $Q(\mathbf{a}, \mathbf{b}) = 0$, so we are on the one dimensional manifold $\mathbf{a} = \pm \mathbf{b}$. Note that, in this case, if $\mathbf{a} = -\mathbf{b}$ then the problem becomes $L = (\|\mathbf{a}\|^2 + \Phi)^2$ and it converges to the minimum $\mathbf{a} = \mathbf{b} = 0$ with $\mu(\mathbf{a}, \mathbf{a}) = (\|\mathbf{a}\|^2 + \Phi)^2 > \Phi^2 > 0$. If, instead, $\mathbf{a} \neq -\mathbf{b}$ and there exists a component $i$ such that $\mathbf{a}_i = \mathbf{b}_i$ instead the components $n_1 < n$ components satisfying $\mathbf{a}_i = -\mathbf{b}_i$ will converge to $\mathbf{a}_i = \mathbf{b}_i = 0$, the the $n - n_1 \geq 1$ other components will converge to the global minimum of $L$ with PL constant given by their norm at initialization $2\sum_{i \text{ s.t. } \mathbf{a}_1 \neq -\mathbf{b}_i} \mathbf{a}_1^2$. This implies that the manifold where the algorithms converge to the saddle is not just of measure zero, but it is just 1 dimensional. This argument proves that

**Proposition 2.** *The loss $L(\mathbf{a}, \mathbf{b})$ equipped with gradient flow is PLAT almost everywhere.*

This proposition demonstrates that $L$ equipped with GF, reliably conforms to the PLAT, showcasing linear convergence, although with a clear differentiation of behavior based on initial conditions and trajectory dynamics. Note that analysis of gradient flow on this model were already present in

the literature: Xu & Ziyin (2024) characterizes the dynamics except for the speed of convergence, and Chatterjee (2022) for a proof technique which, however, only works within a neighborhood of the manifold of minima. Instead, our machinery allows us to complete a proof of global linear convergence very easily, see Appendix D.1.

### 6.4 GRADIENT DESCENT CONVERGES LINEARLY

We show here the analogous version of Proposition 2 for gradient descent now, which is a qualitative version of the first part of Theorem 2.

**Proposition 3.** $L(\mathbf{a}, \mathbf{b})$ *equipped with gradient descent satisfies the PLAT-condition with every step size in* $0 < \eta < \min\left\{ \frac{\sqrt{2}}{\varepsilon}, \frac{2}{\sqrt{\boldsymbol{\lambda}(0)^2 + 4\Phi^2}} \right\}$ *for every initialization* $\mathbf{a}, \mathbf{b} \in \mathbb{R}^n$.

Indeed, if $Q(0) = 0$ then we are in the same cases as above and we anyways have linear convergence, if $Q(0) \neq 0$ then the proof can be found in Appendix D.3. We can thus prove that no matter the initialization and the step size (smaller than a certain value) we have linear convergence somewhere. However, the location of this linear convergence exhibit particular sensitivity to this hyper parameters and in some cases can be a saddle point, see Appendix D.2.

## 7 CONCLUSION

In this paper, we analyzed the gradient descent dynamics of a depth-2 linear neural network, offering a simplified model to explore training behaviors observed in more complex neural networks. Our key technical contributions are:

1. **Linear convergence with large step sizes:** We demonstrated that gradient descent converges at a linear rate to a global minimum, even with larger-than-expected step sizes—up to approximately 2/sharpness. For even larger step sizes, convergence can still occur, but slows down significantly. See Section 4.

2. **Location of convergence:** We characterized the solution reached by gradient descent, showing that it implicitly regularizes the parameter imbalance and sharpness, leading to a lower norm solution compared to gradient flow. Notably, as the step size increases, the implicit regularization effect strengthens, flattening the solution. See Section 3.

The key implications of our results are that

i. **GD always regularizes more than GF:** Gradient descent converges to a solution with lower imbalance than gradient flow, but the imbalance always remains non-zero. The solution is still suboptimal from this perspective. See Section 3.

ii. **GF is not always a good approximation of GD:** We prove that even in a very simple model, gradient flow dynamics are inherently different from gradient descent. In particular, our results can be used as a proof that the common use of GF as a theoretical tool for understanding GD is not always well founded. See Section 3.

iii. **Trade-off Between Speed and Regularization:** Our analysis uncovered a trade-off between the convergence rate and the degree of implicit regularization. See Section 4. Training at the edge of stability, while slower, induces additional regularization, which may be beneficial for generalization. See Section 5.

Our findings thus provide insight into different step sizes affect neural network training dynamics and its potential benefits for regularization in more complex models.

**Future work:** In this work, we studied the model (2) because its simplicity allows for a detailed analysis that leads to the useful conclusions detailed above. However, there are several possible extensions of these results that could lend additional insights. For example, it would be interesting to study the case of vector-valued inputs, deeper models, and non-linear models that use ReLU or other activation functions. In addition, we are interested to know how our results would be impacted by using stochastic gradient descent in rather than exact gradient descent.

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

## A  ON THE OBJECTIVE

**Lemma 4.** *For any* $\mathbf{a}, \mathbf{b}, \{(x_i, y_i)\}_{i=1}^n$,

$$\bar{L}(\mathbf{a}, \mathbf{b}) \;=\; \frac{\sum_{i=1}^n x_i^2}{2n}(\mathbf{a}^\top \mathbf{b} - c)^2 + \textit{Const} \;=\; \left[\frac{1}{n}\sum_i x_i^2\right] L(\mathbf{a}, \mathbf{b}) + \textit{Const}.$$

*where* $c = \frac{\sum_{i=1}^n x_i y_i}{\sum_{i=1}^n x_i^2}$ *and* $\textit{Const} = \frac{1}{2n}\left(\sum_{i=1}^n y_i^2 - \frac{\left(\sum_{i=1}^n x_i y_i\right)^2}{\sum_{i=1}^n x_i^2}\right)$ *does not depend on* $\mathbf{a}, \mathbf{b}$.

*Proof.* Let $\mathbf{x}$ denote the vector whose $i$th entry is $x_i$, and let $\mathbf{y}$ denote the vector whose $i$th entry is $y_i$. Then we can write

$$\bar{L}(\mathbf{a}, \mathbf{b}) = \frac{1}{2n}\left\|\mathbf{a}^\top \mathbf{b}\mathbf{x} - \mathbf{y}\right\|^2 = \frac{1}{2n}\left((\mathbf{a}^\top \mathbf{b})^2\|\mathbf{x}\|^2 - 2\mathbf{a}^\top \mathbf{b}\langle \mathbf{x}, \mathbf{y}\rangle + \|\mathbf{y}\|^2\right)$$

$$= \frac{\|\mathbf{x}\|^2}{2n}\left((\mathbf{a}^\top \mathbf{b})^2 - 2\mathbf{a}^\top \mathbf{b}\frac{\langle \mathbf{x}, \mathbf{y}\rangle}{\|\mathbf{x}\|^2} + \frac{\|\mathbf{y}\|^2}{\|\mathbf{x}\|^2}\right)$$

$$= \frac{\|\mathbf{x}\|^2}{2n}\left(\left(\mathbf{a}^\top \mathbf{b} - \frac{\langle \mathbf{x}, \mathbf{y}\rangle}{\|\mathbf{x}\|^2}\right)^2 + \frac{\|\mathbf{y}\|^2}{\|\mathbf{x}\|^2} - \frac{\langle \mathbf{x}, \mathbf{y}\rangle^2}{\|\mathbf{x}\|^4}\right) \quad (12)$$

Rewriting this in terms of the $x_i$'s and $y_i$'s completes the proof. $\qquad\square$

Note, thus, that all our proofs work on $\bar{L}$, we thus have to rescale $\varepsilon, \boldsymbol{\lambda}, Q, \eta$ accordingly. Precisely,

$$\boldsymbol{\lambda} \curvearrowleft \left[\frac{1}{n}\sum_i x_i^2\right]\boldsymbol{\lambda}, \quad Q \curvearrowleft \left[\frac{1}{n}\sum_i x_i^2\right]Q, \quad \varepsilon \curvearrowleft \left[\frac{1}{n}\sum_i x_i^2\right]\varepsilon, \quad \text{and} \quad \eta \curvearrowleft \left[\frac{1}{n}\sum_i x_i^2\right]\eta. \tag{13}$$

Analogously, note that if $\Phi < 0$ nothing changes in the analysis of the dynamics. When $\mathbf{a} \neq -\mathbf{b}$ just change $\mathbf{a}$ to $-\mathbf{a}$ and apply the same analysis as before.

## B  FROM THE RESIDUALS TO THE LOSS

First note that if $\varepsilon$ converges exponentially to zero, then loss $L$ converges exponentially to its minimum.

**Lemma 5.** *Assume $|\varepsilon(k)|$ converges linearly fast with rate $(1-\eta\mu) < 1$. Then $L$ converges linearly fast with rate $(1-\eta\mu)^2$. In particular, let $\delta > 0$, the loss $L$ is smaller than $\delta$ in a number of steps $t$ that satisfies*

$$t \leq \frac{\log L_0 - \log(\delta)}{\eta\mu}.$$

Indeed note that for how we defined $\varepsilon$ we have that $L = \varepsilon^2$, thus $L(k+1) = |\varepsilon(k+1)| \leq (1-\eta\mu)|\varepsilon(k)|^2$. Note that this lemma allows us to deal with the convergence of $\varepsilon$ instead of the convergence of $L$ and infer the convergence of $L$. Indeed, if the residuals $\varepsilon$ converge linearly with rate $(1-\eta\mu) < 1$, then the time it takes to converge is such that $\sqrt{\delta} \geq (1-\eta\mu)^t\sqrt{L_0}$ which is

$$t \leq \frac{\log L_0 - \log(\delta)}{-\log(1-\eta\mu)} \leq \frac{\log L_0 - \log(\delta)}{\eta\mu}. \tag{14}$$

From now on we will deal with convergence of residuals only.

## C  BOUNDING THE FINAL SHARPNESS

### C.1  SIZE OF $\boldsymbol{\lambda}$ FOR GRADIENT FLOW

Note that we can characterize the norm $\boldsymbol{\lambda}_\infty$ found by gradient flow by noticing that

**Lemma 6.** *Along the gradient flow trajectory, the following quantity is conserved*

$$\boldsymbol{\lambda}^2 - 8\varepsilon(\varepsilon + \Phi) + 4\varepsilon^2.$$

*Proof.* The gradient flow dynamics are described by

$$\begin{bmatrix}\dot{\mathbf{a}}\\\dot{\mathbf{b}}\end{bmatrix} = -\nabla L(\mathbf{a}, \mathbf{b}) = -\varepsilon\begin{bmatrix}\mathbf{b}\\\mathbf{a}\end{bmatrix} \tag{15}$$

First, we compute

$$\dot{\boldsymbol{\lambda}} = \frac{d}{dt}\left[\|\mathbf{a}\|^2 + \|\mathbf{b}\|^2\right] \tag{16}$$

$$= 2\langle\mathbf{a}, \dot{\mathbf{a}}\rangle + 2\left\langle\mathbf{b}, \dot{\mathbf{b}}\right\rangle \tag{17}$$

$$= -2\varepsilon\langle\mathbf{a}, \mathbf{b}\rangle - 2\varepsilon\langle\mathbf{b}, \mathbf{a}\rangle \tag{18}$$

$$= -4\varepsilon(\varepsilon + \Phi) \tag{19}$$

and

$$\dot{\varepsilon} = \frac{d}{dt}[\langle\mathbf{a}, \mathbf{b}\rangle - \Phi] \tag{20}$$

$$= \left\langle\mathbf{a}, \dot{\mathbf{b}}\right\rangle + \langle\dot{\mathbf{a}}, \mathbf{b}\rangle \tag{21}$$

$$= -\varepsilon\|\mathbf{a}\|^2 - \varepsilon\|\mathbf{b}\|^2 \tag{22}$$

$$= -\boldsymbol{\lambda}\varepsilon \tag{23}$$

Finally, straightforward calculation confirms:

$$\frac{d}{dt}\left[\boldsymbol{\lambda}^2 - 8\varepsilon(\varepsilon + \Phi) + 4\varepsilon^2\right] = 2\boldsymbol{\lambda}\dot{\boldsymbol{\lambda}} - 8\varepsilon\dot{\varepsilon} - 8\dot{\varepsilon}(\varepsilon + \Phi) + 8\varepsilon\dot{\varepsilon} \tag{24}$$

$$= 2\boldsymbol{\lambda}\dot{\boldsymbol{\lambda}} - 8\dot{\varepsilon}(\varepsilon + \Phi) \tag{25}$$

$$= 2\boldsymbol{\lambda}(-4\varepsilon(\varepsilon + \Phi)) - 8(-\boldsymbol{\lambda}\varepsilon)(\varepsilon + \Phi) \tag{26}$$

$$= 0 \tag{27}$$

which completes the proof. $\qquad\square$

**Lemma 7.** $\boldsymbol{\lambda}(t)$ *along the whole GF trajectory satisfies*

$$\boldsymbol{\lambda}(\infty) \quad \leq \quad \sqrt{\boldsymbol{\lambda}(0)^2 + 4\Phi^2}.$$

*Proof.* Note that

$$\boldsymbol{\lambda}(\infty) = \boldsymbol{\lambda}(0)^2 - 8\varepsilon(0)(\varepsilon(0) + \Phi) + 4\varepsilon(0)^2.$$

Note that the maximum over $\lambda = \boldsymbol{\lambda}(0)$ of $-8\varepsilon(0)(\varepsilon(0) + \Phi) + 4\varepsilon(0)^2$ is

$$4 \max_{\boldsymbol{\lambda}=\boldsymbol{\lambda}(0)} -\varepsilon(\varepsilon + 2\Phi) \tag{28}$$

Is at $\varepsilon = -\Phi$. This implies that for all the points with fixed $\boldsymbol{\lambda}$ the one with highest $\boldsymbol{\lambda}^2 - 8\varepsilon(\varepsilon + \Phi) + 4\varepsilon^2$ is the one with $\varepsilon = -\Phi$. Whatever was the initialization with a certain fixed scale, the solution found will have lambda smaller than $\boldsymbol{\lambda}^2 - 8\varepsilon(\varepsilon + \Phi) + 4\varepsilon^2$, thus of $\sqrt{\boldsymbol{\lambda}(0)^2 + 4\Phi^2}$. Next note that $\lambda$ has positive derivative only when $\varepsilon \in [-\Phi, 0)$. This implies that the sup for $\lambda$ along the trajectory is either initialization or the solution. $\qquad\square$

### C.2 Size of $\boldsymbol{\lambda}$ for Gradient Descent

Surprisingly, we show here that if switch to gradient descent the quantity $\boldsymbol{\lambda}^2 - 8\varepsilon(\varepsilon + \Phi) + 4\varepsilon^2$ actually decreases to the second order in $\eta$.

**Lemma 8.** *One step of gradient descent trajectory with step size $\eta > 0$, induces the following change in the quantity $\boldsymbol{\lambda}^2 - 8\varepsilon(\varepsilon + \Phi) + 4\varepsilon^2$:*

$$\boldsymbol{\lambda}_1^2 - 8\varepsilon_1(\varepsilon_1 + \Phi) + 4\varepsilon_1^2 \quad \curvearrowleft \quad \boldsymbol{\lambda}^2 - 8\varepsilon(\varepsilon + \Phi) + 4\varepsilon^2 \; - \; 2\eta^2\varepsilon^2 Q^2(1 - \eta^2\varepsilon^2).$$

*Proof.* Note that

$$\boldsymbol{\lambda}_1^2 = \left((\mathbf{a} - \eta\varepsilon\mathbf{b})^2 + (\mathbf{b} - \eta\varepsilon\mathbf{a})^2\right)^2$$

$$= \left(\boldsymbol{\lambda}(1 + \eta^2\varepsilon^2) - 4\eta\varepsilon(\varepsilon + \Phi)\right)^2 \tag{29}$$

$$= \boldsymbol{\lambda}^2 - 8\eta\varepsilon\boldsymbol{\lambda}(\varepsilon + \Phi) + 2\eta^2\varepsilon^2\boldsymbol{\lambda}^2 + \eta^4\varepsilon^4\boldsymbol{\lambda}^2 + 16\eta^2\varepsilon^2(\varepsilon + \Phi)^2 - 8\eta^3\varepsilon^3\boldsymbol{\lambda}(\varepsilon + \Phi).$$

Analogously

$$4\varepsilon_1^2 = 4\left(\varepsilon(1 - \eta\boldsymbol{\lambda}) + \eta^2\varepsilon^2(\varepsilon + \Phi)\right)^2$$

$$= 4\varepsilon^2 - 8\eta\varepsilon^2\boldsymbol{\lambda} + 4\eta^2\varepsilon^2\boldsymbol{\lambda}^2 + 8\eta^2\varepsilon^3(\varepsilon + \Phi) - 8\eta^3\varepsilon^3\boldsymbol{\lambda}(\varepsilon + \Phi) + 4\eta^4\varepsilon^4(\varepsilon + \Phi)^2, \tag{30}$$

and

$$-8\varepsilon_1(\varepsilon_1 + \Phi) = -8\left(\varepsilon(1 - \eta\boldsymbol{\lambda}) + \eta^2\varepsilon^2(\varepsilon + \Phi)\right)\left(\Phi + \varepsilon(1 - \eta\boldsymbol{\lambda}) + \eta^2\varepsilon^2(\varepsilon + \Phi)\right)$$

$$= -8\varepsilon(\varepsilon + \Phi) + 8\eta\varepsilon\boldsymbol{\lambda}(2\varepsilon + \Phi) - 8\eta^2\varepsilon^2\boldsymbol{\lambda}^2 \tag{31}$$

$$- 8\eta^2\varepsilon^2(\varepsilon + \Phi)\left((2\varepsilon + \Phi) - 2\eta\boldsymbol{\lambda}\varepsilon + \eta^2\varepsilon^2(\varepsilon + \Phi)\right).$$

This (Lemma 6) implies that the monomials of degree 1 in $\eta$ zeroes out, the monomial of degree 3 zeroes out too:

$$\eta^3\varepsilon^3 \cdot \left(-8\boldsymbol{\lambda}(\varepsilon + \Phi) - 8\boldsymbol{\lambda}(\varepsilon + \Phi) + 16\boldsymbol{\lambda}(\varepsilon + \Phi)\right) \quad = \quad 0. \tag{32}$$

The monomials of degree 2 in $\eta$ are

$$\eta^2 \varepsilon^2 \cdot \left( \boldsymbol{\lambda}^2(2+4-8) + (\varepsilon + \Phi)^2(16-8) + \varepsilon(\varepsilon + \Phi)(8-8) \right) \;=\; -2\eta^2 \varepsilon^2 (\boldsymbol{\lambda}^2 - 4(\varepsilon + \Phi)^2). \tag{33}$$

This is exactly equal to

$$-2\eta^2 \varepsilon^2 Q^2.$$

Analogously the monomial of degree 4 in $\eta$ is

$$2\eta^4 \varepsilon^4 Q^2$$

which completes the proof. $\qquad\square$

**Lemma 9.** *Let $\eta < 1/|\epsilon(0)|$ and assume $|\epsilon(t)|$ is monotonically decreasing, along the GD trajectory*

$$\boldsymbol{\lambda}(t) \;\leq\; \sqrt{\boldsymbol{\lambda}(0)^2 + 4\Phi^2}.$$

The proof follows as the one of Lemma 7 by exchanging the equalities given by Lemma 6 with the inequalities given by Lemma 8.

**Definition 3** (Maximal Sharpness $\bar{\boldsymbol{\lambda}}$)**.** We denote by $\bar{\boldsymbol{\lambda}}$ and we call maximal sharpness the value

$$\bar{\boldsymbol{\lambda}} \;:=\; \sqrt{(\|\mathbf{a}\|^2 + \|\mathbf{b}\|^2)^2 + 4\Phi^2}.$$

# D    SPEED OF CONVERGENCE

## D.1    CONTINUOUS DYNAMICS: PROOF OF PROPOSITION 2

In the case of gradient flow the pairs $(\mathbf{a}, \mathbf{b})$ along the trajectory satisfy a PL condition with $\mu(\mathbf{a}, \mathbf{b}) = \|\mathbf{a}\|^2 + \|\mathbf{b}\|^2$, indeed note that $L(\mathbf{a}, \mathbf{b})$ satisfies

$$\left(\mathbf{a}^\top \mathbf{b} - \Phi\right)^2 \left(\|\mathbf{a}\|^2 + \|\mathbf{b}\|^2\right) \;=\; \|\nabla L(\mathbf{a}, \mathbf{b})\|^2 \;=\; \mu(t) \cdot L(\mathbf{a}, \mathbf{b}).$$

Note that for all $i$ the quantity $Q_i = \mathbf{a}_i^2 - \mathbf{b}_i^2$ is conserved along the trajectory, indeed

$$\frac{d}{dt}\left(\mathbf{a}_i(t)^2 - \mathbf{b}_i(t)^2\right) \;=\; 2\varepsilon(\mathbf{a}_i \mathbf{b}_i - \mathbf{a}_i \mathbf{b}_i) \;=\; 0.$$

Thus we have that $Q(0) \neq 0$ is a lower bound to $\mu$ along the whole trajectory, we thus proved that

**Lemma 10.** *Let $\mathbf{a}, \mathbf{b}$ such that $Q \neq 0$. The gradient flow starting from $a, b$ converges exponentially fast with rate at least $Q$ to the point $\mathbf{a}(\infty), \mathbf{b}(\infty)$ which satisfies that (i) $\mathbf{a}(\infty)^\top \mathbf{b}(\infty) = \Phi$ and (ii) for all $i$ that $Q_i(0) = Q_i(\infty)$ and $sign\big(\mathbf{a}_i(\infty) - \mathbf{b}_i(\infty)\big) = sign\big(\mathbf{a}_i(0) - \mathbf{b}_i(0)\big)$.*

This lemma and the observation of what happens in the case of $Q = 0$ in Section D.2, prove Proposition 2.

## D.2    COMMENT ON PROPOSITIONS 2 AND 3

Note that for a fixed initialization where $Q \neq 0$, if $\eta$ is such that there exists a step $k$ along the trajectory where $\eta \cdot (\mathbf{a}^\top \mathbf{b} - \Phi) = 1$ exactly, convergence happen to $\mathbf{a} = \mathbf{b} = 0$ instead of the global minimum. Indeed, in this case, on the next step we have $\mathbf{a}(k+1) = -\mathbf{b}(k+1) = \mathbf{a}(k) - \mathbf{b}(k)$. This implies that when $Q \neq 0$, for almost every $\eta$ in the allowed range we have $Q \neq 0$ along the whole trajectory, and as we prove, linear convergence to a global minimum.

We characterize below what happens in the case in which $Q = 0$ at some point along the trajectory.

For both GD and GF if at a certain point during the training (or at initialization) $\mathbf{a}$ and $\mathbf{b}$ are such that $Q(\mathbf{a}, \mathbf{b}) = 0$, then we are on the one dimensional manifold in which for every neuron $i$ we have $\mathbf{a}_i = \pm \mathbf{b}_i$.

- If $\mathbf{a} = -\mathbf{b}$ then the problem becomes $L = (\|\mathbf{a}\|^2 + \Phi)^2$ and it converges to the minimum $\mathbf{a} = \mathbf{b} = 0$ of the modified loss $\tilde{L} = \|a\|^2$. The gradient is such that

$$\|\nabla L(\mathbf{a},\mathbf{b})\|^2 \;=\; \left(\|\mathbf{a}\|^2 + \Phi\right)^2 \cdot 2\|\mathbf{a}\|^2 \;=\; \mu(t) \cdot \tilde{L}(\mathbf{a},\mathbf{b}).$$

  with $\mu(\mathbf{a},\mathbf{a}) = 2(\|\mathbf{a}\|^2 + \Phi)^2 \geq 2\Phi^2 > 0$. Thus restricted to the manifold where the trajectory lies, we have a function satisfying the PL condition with $\mu \geq 2\Phi^2 > 0$. In this case both GD and GF converge linearly fast to the minimum along this manifold, i.e., the saddle point at the origin.

- If $\mathbf{a} \neq -\mathbf{b}$ and there exists a component $i$ such that $\mathbf{a}_i = \mathbf{b}_i$ instead the components $n_1 < n$ components satisfying $\mathbf{a}_i = -\mathbf{b}_i$ will converge to $\mathbf{a}_i = \mathbf{b}_i = 0$, the the $n - n_1 \geq 1$ other components will converge to the global minimum of $L$ with PL constant given by their norm at initialization $2\sum_{i \text{ s.t. } \mathbf{a}_1 \neq -\mathbf{b}_i} \mathbf{a}_1^2$. This implies that in this case we have convergence to 0 for the neurons in which $\mathbf{a}_i = \mathbf{b}_i$ and the dynamics is as described in the rest of the manuscript for the other neurons in which $\mathbf{a}_i = -\mathbf{b}_i$.

This implies that the manifold where the algorithms converge to the saddle is not just of measure zero, but it is precisely $\mathbf{a} = -\mathbf{b}$. Even in this case, we have linear convergence to the saddle, when the learning rate is smaller than $2/\lambda$. In all the other cases, if $Q = 0$, we have a sub network where $\mathbf{a} = \mathbf{b} \neq 0$, thus the loss satisfies

$$\|\nabla L(\mathbf{a},\mathbf{b})\|^2 \;=\; \left(\|\mathbf{a}\|^2 - \Phi\right)^2 \cdot 2\|\mathbf{a}\|^2 \;=\; \mu(t) \cdot \tilde{L}(\mathbf{a},\mathbf{b}).$$

with PL-condition $2\|\mathbf{a}\|_2^2$, which is positive and bounded below by $2\|\mathbf{a}(0)\|_2^2$ if initialized in Area B, and by $2\Phi^2 > 0$ if we initialized in Area A.

We thus have linear convergence either to the saddle at the origin or to a global minimum for $Q = 0$. In the rest we abnalyze the case $Q \neq 0$.

### D.3 LOWER BOUND TO $\mu(t)$ IN THE DISCRETE CASE.

Note that the derivative in time of $\mu(\mathbf{a}(t), \mathbf{b}(t))$ is

$$\dot{\mu} \;=\; -4 \left[\frac{1}{n}\sum x_i^2\right]^2 \left(\mathbf{a}^\top \mathbf{b} - \Phi\right) \mathbf{a}^\top \mathbf{b}. \tag{34}$$

It thus decreases when $\mathbf{a}^\top \mathbf{b} > \Phi$ and when $\mathbf{a}^\top \mathbf{b} < 0$ and $\mathbf{a}^\top \mathbf{b} < \Phi$, it grows when $\mathbf{a}^\top \mathbf{b} > 0$ and $\mathbf{a}^\top \mathbf{b} < \Phi$. This means that

**Area A)** When $\mathbf{a}(0)^\top \mathbf{b}(0) > \Phi$, in Area A of Figure 2, we can bound

$$\mu(t) \geq \inf_{\mathbf{a}^\top \mathbf{b} > \Phi} \left[\frac{1}{n}\sum x_i^2\right] (\|\mathbf{a}\|^2 + \|\mathbf{b}\|^2) = 2\Phi.$$

Thus in this area we have that $2\Phi \leq \boldsymbol{\lambda} \leq \bar{\boldsymbol{\lambda}}$.

**Area B)** When $\mathbf{a}(0)^\top \mathbf{b}(0) > 0$ and $\mathbf{a}(0)^\top \mathbf{b}(0) < \Phi$, in Area B of Figure 2, we can bound

$$\mu(t) \geq \mu(0) = \left[\frac{1}{n}\sum x_i^2\right] (\|\mathbf{a}(0)\|^2 + \|\mathbf{b}(0)\|^2).$$

Thus in this area we have that $\boldsymbol{\lambda}(0) \leq \boldsymbol{\lambda} \leq 2\Phi \leq \bar{\boldsymbol{\lambda}}$, where $\boldsymbol{\lambda}(0) \geq Q_0 > 0$ is the norm of the first step in this area, when $Q \neq 0$.

Note that this implies that our loss equipped with gradient descent is PLAT in Area A and Area B.

**Area C)** When $\mathbf{a}(0)^\top \mathbf{b}(0) < 0$, in Area C of Figure 2, the residuals decreases until $\mathbf{a}^\top \mathbf{b} = 0$. Thus the lowest point for $Q$ will be at the step $\tau$ that is the first step in which $\mathbf{a}^\top \mathbf{b} \geq 0$. This implies that the quantity $\boldsymbol{\lambda}$ will be at its minimum either at time $\tau$ or $\tau - 1$

$$\mu(t) \geq \min\{\mu(\tau - 1), \mu(\tau)\} \quad \text{where } \tau = \min_{t\in\mathbb{N}}\{\mathbf{a}(t)^\top \mathbf{b}(t) > 0\}.$$

In particular $\mu(t) \geq \min\{\mu(\tau - 1), \mu(\tau)\} \geq Q(\tau_1)$, we need to show that when $Q \neq 0$ then $Q(\tau_1) \neq 0$. Thus in this area we will prove in the next section that we have that $Q(\tau_1) \leq \boldsymbol{\lambda} \leq 2\Phi \leq \bar{\boldsymbol{\lambda}}$.

This concludes the argument for all the cases except for $\mathbf{a}(0)^\top \mathbf{b}(0) < 0,$. We will now bound $\left|Q_i(\tau_1)\right|$ in terms of $\left|Q_i(0)\right|$, the learning rate $\eta > 0$, and $\mathbf{a}(0)^\top \mathbf{b}(0)$.

# E  LOWER BOUND ON $\mu$ IN AREA C

We prove in this section that

    1. The loss equipped with gradient descent is PLAT also in Area C.

    2. That GD escapes Area C very quickly, precisely see Proposition 4.

Precisely, in Subsection E.1 we show the two points above for $\eta \geq \frac{1}{2|\varepsilon|}$ at initialization. In Subsection E.2 we show it for the case of smaller step size, precisely, we show there the following Proposition:

**Proposition 4.** *If* $\mathbf{a}(0)^\top \mathbf{b}(0) < 0$ *and* $\Phi > 0$, *there exists* $\tau$ *such that* $a(\tau)^\top b(\tau) > 0$ *and*

$$2\sqrt{\eta}\Phi \quad < \quad Q(\tau) \quad = \quad O\left(\exp(-\eta)\right) \cdot Q(0).$$

    • *If* $\sum_i |\mathbf{a}_i^2(0) - \mathbf{b}_i^2(0)| \geq 2|\mathbf{a}(0)^\top \mathbf{b}(0)|$ *then* $\tau \leq O\left(\eta^{-1}\right)$ *and*

    • *If* $\sum_i |\mathbf{a}_i^2(0) - \mathbf{b}_i^2(0)| < 2|\mathbf{a}(0)^\top \mathbf{b}(0)|$ *then* $\tau \leq O\left(\frac{1}{\log(1+2\eta)}\right)$.

*Analogously if* $\mathbf{a}(0)^\top \mathbf{b}(0) > 0$ *and* $\Phi < 0$, *after the same* $\tau$, *we have* $\mathbf{a}(\tau)^\top \mathbf{b}(\tau) < 0$ *and* $\sum_i |\mathbf{a}_i^2(\tau) - \mathbf{b}_i^2(\tau)|$ *is the same as above.*

## E.1  BIGGER STEP SIZE

Note that if

$$\eta \quad \geq \quad \frac{1}{|\varepsilon|} \frac{|\mathbf{a}^\top \mathbf{b}|}{\|\mathbf{a}\|^2 + \|\mathbf{b}\|^2} \quad \geq \quad \frac{1}{2|\varepsilon|}$$

then in the next step we are landing directly in Area $B$ or $A$. Precisely, this implies that $\mathbf{a}^\top \mathbf{b} > 0$ after one step. In these cases gradient descent makes one step in Area C and leaves, so the convergence analysis continues with the ones in Areas B and A.

## E.2  SMALLER STEP SIZES

The difficult case to deal with analytically is the one where the dynamics stays in Area C for long.

We compute here a lower bound on $|Q_i(\tau)|$. The idea here is that the residuals $\mathbf{a}^\top \mathbf{b} - \Phi$ will converge as $\exp(-\eta t)$ and the quantity $Q_i(t)$ at most as $\exp(-\eta^2 t)$, thus $\mathbf{a}(t)^\top \mathbf{b}(t)$ crosses 0 before $|Q_i(t)|$ gets too small.

Note that at every step of gradient descent we have the following updates on the following quantities

$$\mathbf{a}(t+1)^\top \mathbf{b}(t+1) - \Phi \quad = \quad \left(1 - \eta\left(\|\mathbf{a}(t)\|^2 + \|\mathbf{b}(t)\|^2\right)\right)\left(\mathbf{a}(t)^\top \mathbf{b}(t) - \Phi\right)$$
$$+ \eta^2 \underbrace{\left(\mathbf{a}(t)^\top \mathbf{b}(t) - \Phi\right)^2 \mathbf{a}(t)^\top \mathbf{b}(t)}_{positive} \tag{35}$$

and

$$\mathbf{a}_i(t+1)^2 - \mathbf{b}_i(t+1)^2 \quad = \quad \left(1 - \eta^2\left(\mathbf{a}(t)^\top \mathbf{b}(t) - \Phi\right)^2\right)\left(\mathbf{a}_i(t)^2 - \mathbf{b}_i(t)^2\right). \tag{36}$$

Thus we have that

$$\mathbf{a}(t+1)\mathbf{b}(t+1) - \Phi \quad > \quad \left(1 - \eta\left(\|\mathbf{a}(t)\|^2 + \|\mathbf{b}(t)\|^2\right)\right)\left(\mathbf{a}(t)^\top \mathbf{b}(t) - \Phi\right) \tag{37}$$

when $\mathbf{a}(t)^\top \mathbf{b}(t) < 0$.

**Bounding Sequences.** We define here two coupled sequences which serve as bounds to the evolution of $\varepsilon$ and $Q$ along then trajectory. We study their behavior and we infer bounds on the behavior of our system.

**Definition 4.** Let $\mathbf{a}(0), \mathbf{b}(0) \in \mathbb{R}^n$. Define $\eta < \min\left\{\frac{1}{2|\varepsilon|}, \frac{2}{\lambda}\right\}$. Define the sequence $\{z_k, w_k\}_{k=0}^{\infty}$ such that $z_0 = \varepsilon(0) < -\Phi$, $w_0 = Q_0 > 0$, and for all $k \in \mathbb{N}$ we have

$$
\begin{aligned}
z_{k+1} &= \left(1 - \eta \max\left\{w_k, -2z_k - 2\Phi\right\}\right) z_k \\
w_{k+1} &= \left(1 - \eta^2 z_k^2\right) w_k.
\end{aligned}
\tag{38}
$$

Define $\tau_1 := \min_{k \in \mathbb{N}}\{z_k > -\Phi\}$.

Note that we have

**Lemma 11** (Bounding with the sequences). *For all $1 \le k < \tau_1$ such that $z_k < 0$ we have*

$$
z_k \le \varepsilon_k \quad \text{and} \quad w_k < Q_k.
$$

*Moreover, $w_k, -z_k \ge 0$ are strongly monotone decreasing for $k < \tau_1$ and $a(\tau_1)^\top b(\tau_1) > 0$, thus for all $k \le \tau_1$ we have $\eta < 2/\max\{-2z_k, w_k\}$.*

*Proof.* Note that this is the case for $k = 0$. As for the inductive step, Eq. 37, Cauchy-Schwartz inequality, and Eq. 38 estblish the first point. Note that $z_{\tau_1 - 1} + \Phi < a(\tau_1 - 1)b(\tau_1 - 1) < 0$, then the first point and the definition of $\tau$ imply that $0 < z_{\tau_1} + \Phi < a(\tau_1)^\top b(\tau_1)$. Note that after the first step $w_1 < w_0$ and $z_1 > z_0$ since Cauchy-Schwartz implies that $\eta < 2/\max\{-2z_0, w_0\}$. Inductively, for all $i$ we have $\eta < 2/\max\{-2z_i, w_i\}$, thus fact that $z_i < 0$ for all $k < \tau_1$ implies that $w_{i+1} < w_i$ is strongly monotonically decreasing, that $z_{i+1} > z_i$ is strongly monotonically increasing, and that $\eta < 2/\max\{-2z_{i+1}, w_{i+1}\}$. $\qquad\square$

As explained before, for all $t$ we have $\mu(t) \ge \max\{\mu(\tau_1), \mu(\tau_1 - 1)\}$ and $\mu(t) \ge \sum_i |a_i^2(t) - b_i^2(t)| \ge w_t \ge w_{t+1}$. Thus for all $t$ we have $\mu(t) \ge w_{\tau_1}$. This and the lemma above show that

**Lemma 12.** *We have that $a(\tau)^\top b(\tau) > -\Phi$ and for all $t \in \mathbb{N}$ we have $\mu(t) \ge w_\tau$.*

**Behavior of the sequence: Case 1.** We assume in this paragraph that $w_0 \ge -2z_0 - 2\Phi$. We characterize $\tau$ and $Q(\tau)$ in this case.

**Lemma 13** (Rate of convergence 1 - Sequence.). *If $w_0 \ge -2z_0 - 2\Phi$, define $c_1 := \frac{w_0}{2} - \frac{\sqrt{w_0(w_0 - 4\eta z_0^2)}}{2} > 0$, then*

$$
c_1 \quad < \quad w_{\tau_1} \quad < \quad w_0 - \eta^2 \left(\Phi\right)^2 \frac{\left(\mathbf{a}(0)^\top \mathbf{b}(0)\right)^2}{w_0}
\tag{39}
$$

*and*

$$
\frac{1}{\eta(w_0)^{3/2}} \quad < \quad \tau_1 \quad < \quad \frac{\mathbf{a}(0)^\top \mathbf{b}(0)}{\eta c_1^{3/2}} \left(\Phi\right)^{-1} + 1.
\tag{40}
$$

*Proof.* Note that for all $k < \tau_1$ we have

$$
\frac{(z_{k+1} - z_k)^2}{w_{k+1} - w_k} = \frac{\eta^2 w_k^2 z_k^2}{-\eta^2 z_k^2 w_k} = -w_k.
\tag{41}
$$

Note that $z_{\tau_1 - 1} - z_0 < |\mathbf{a}(0)^\top \mathbf{b}(0)| \le z_{\tau_1} - z_0$. We thus obtain that

$$
\mathbf{a}(0)^\top \mathbf{b}(0) \sim z_0 - z_{\tau_1} = \sum_{k=0}^{\tau_1 - 1} z_k - z_{k+1} = \sum_{k=0}^{\tau_1 - 1} \sqrt{w_k - w_{k+1}} \cdot w_k = \eta \sum_{k=0}^{\tau_1 - 1} z_k(w_k)^{3/2}
\tag{42}
$$

This implies that

$$
\eta(\tau_1 - 1)\Phi(w_{\tau - 2})^{3/2} < \mathbf{a}(0)^\top \mathbf{b}(0) \le \eta\tau_1 z_0(w_0)^{3/2}.
\tag{43}
$$

Next we proceed bounding $w_{\tau_1}$ so that we can bound $\tau_1$. Note that the fact that $z_k < -\Phi < 0$ for all $k < \tau_1$ and Sedrakyan's lemma imply that

$$w_0 - w_{\tau_1} = \sum_{i=0}^{\tau_1 - 1} w_k - w_{k+1} = \sum_{k=0}^{\tau_1 - 1} \frac{(z_{k+1} - z_k)^2}{w_k} > \frac{(z_{\tau_1} - z_0)^2}{\sum_{i=0}^{\tau_1 - 1} w_k}$$

$$> \frac{\left(\mathbf{a}(0)^\top \mathbf{b}(0)\right)^2}{\sum_{i=0}^{\tau_1 - 1} (1 - \eta^2 z_{\tau_1 - 1}^2) w_0} = \eta^2 z_{\tau_1}^2 \frac{\left(\mathbf{a}(0)^\top \mathbf{b}(0)\right)^2}{w_0} \tag{44}$$

$$> \frac{\mathbf{a}(0)^\top \mathbf{b}(0)}{\tau_1 w_0}.$$

And this implies that

$$w_{\tau_1} = w_0 + (w_{\tau_1} - w_0) < w_0 - \eta^2 \left(\Phi\right)^2 \frac{\left(\mathbf{a}(0)^\top \mathbf{b}(0)\right)^2}{w_0} \tag{45}$$

Moreover, we have

$$w_0 - w_{\tau_1} = \sum_{k=0}^{\tau_1 - 1} w_k - w_{k+1} = \eta^2 \sum_{i=0}^{\tau_1 - 1} z_k^2 w_k < \eta^2 \sum_{i=0}^{\tau_1 - 1} z_k^2 (1 - \eta^2 z_{\tau_1}^2)^k w_0$$

$$< \eta^2 z_0^2 w_0 \sum_{i=0}^{\tau_1 - 1} (1 - \eta w_{\tau_1})^k (1 - \eta^2 z_{\tau_1}^2)^k < \eta z_0^2 w_0 \frac{1}{w_{\tau_1} + \eta z_{\tau_1}^2 - \eta^2 w_{\tau_1} z_{\tau_1}^2} \tag{46}$$

$$< \eta \frac{z_0^2 w_0}{w_{\tau_1}}.$$

This implies with $0 < w_{\tau_1} < w_0$ that $w_{\tau_1}(w_0 - w_{\tau_1}) < \eta z_0^2 w_0$, thus

$$w_{\tau_1}^2 - w_0 w_{\tau_1} + \eta z_0^2 w_0 > 0. \tag{47}$$

Note that $w_0 > -2z_0$ and $\eta < -2/z_0$ implies that $w_0 \geq \eta z_0^2$ then, solving, we obtain

$$0 < c_1 := \frac{w_0}{2} - \frac{\sqrt{w_0(w_0 - 4\eta z_0^2)}}{2} < w_{\tau_1} < w_0 - \eta^2 \left(\Phi\right)^2 \frac{\left(\mathbf{a}(0)^\top \mathbf{b}(0)\right)^2}{w_0} \tag{48}$$

$w_{\tau_1} > \frac{w_0}{2} - \frac{\sqrt{w_0(w_0 - 4\eta z_0^2)}}{2}$ Thus, opportunely bounding $w_{\tau-2}$ we obtain

$$\eta(\tau_1 - 1)\Phi c_1^{3/2} < \mathbf{a}(0)^\top \mathbf{b}(0). \tag{49}$$

That we can reorganize as

$$\tau_1 < \frac{\mathbf{a}(0)^\top \mathbf{b}(0)}{\eta c_1^{3/2}} \left(\Phi\right)^{-1} + 1. \tag{50}$$

Thus

$$\frac{1}{\eta(w_0)^{3/2}} \leq \tau_1 \leq \frac{\mathbf{a}(0)^\top \mathbf{b}(0)}{\eta c_1^{3/2}} \left(\Phi\right)^{-1} + 1. \tag{51}$$

$\square$

**Lemma 14** (Rate of convergence 1.). *If* $\sum_i |\mathbf{a}_i^2(0) - \mathbf{b}_i^2(0)| \geq -2\mathbf{a}(0)^\top \mathbf{b}(0) > 0$, *define* $c_1 := \frac{w_0}{2} - \frac{\sqrt{w_0(w_0 - 4\eta z_0^2)}}{2} > 2\sqrt{\eta}\Phi$ *as above, then*

$$c_1 < \sum_i |a_i^2(\tau_1) - b_i^2(\tau_1)| < \sum_i |\mathbf{a}_i^2(0) - \mathbf{b}_i^2(0)| - \eta^2 \left(\Phi\right)^2 \frac{\left(\mathbf{a}(0)^\top \mathbf{b}(0)\right)^2}{\sum_i |\mathbf{a}_i^2(0) - \mathbf{b}_i^2(0)|}, \tag{52}$$

$$\mathbf{a}(0)^\top \mathbf{b}(0) > 0, \tag{53}$$

*and*

$$\frac{1}{\eta \left(\sum_i |\mathbf{a}_i^2(0) - \mathbf{b}_i^2(0)|\right)^{3/2}} < \tau_1 < \frac{\mathbf{a}(0)^\top \mathbf{b}(0)}{\eta c_1^{3/2}} \left(\Phi\right)^{-1} + 1. \tag{54}$$

*Proof.* One bound comes from Lemma 13 and Lemma 11. The other one comes by just following the proof of Lemma 13. $\square$

**Behavior of the sequence: Case 2.** We assume in this paragraph that $w_0 \leq -2z_0 - 2\Phi$. We characterize $\tau_2$ and $Q(\tau_2)$ such that after time $\tau_2$ the sequences are in the Case 1 analyzed above.

**Lemma 15** (Rate of convergence 2 - Sequence.). *Let $c_2 := \left| w_0/\mathbf{a}(0)^\top \mathbf{b}(0) \right|$. If $w_0 < -2z_0 - 2\Phi$ then by time $\tau_2$ as below we are under the assumptions of Lemma 13, so we have $w_{\tau_2} > -2z_{\tau_2} - 2\Phi$ and we have*

$$\frac{|c_2|}{2\eta|z_0|} \quad < \quad \tau_2 \quad < \quad \frac{|c_2|}{\log\left(1 + 2\eta\Phi\right)} + 1, \tag{55}$$

$$w_0(1 - \eta^2 z_0^2) \exp\left(-\frac{\eta|c_2|}{2} \frac{z_0^2 \sum x_i^2}{\sum x_i y_i}\right) \quad < \quad w_{\tau_2} \quad < \quad w_0 \exp\left(-\frac{\eta|c_2|}{2} \frac{\left[\sum x_i y_i\right]^2}{z_0 \left[\sum x_i^2\right]^2}\right), \tag{56}$$

$$\left(z_0 + \Phi\right)\left(1 - 2\eta|z_0|\right) \exp\left(-|c_2| \frac{|z_0| \sum x_i^2}{\sum x_i y_i}\left(1 - \eta\Phi\right)^{-1}\right) - \Phi \quad < \quad z_{\tau_2}, \tag{57}$$

*and*

$$z_{\tau_2} \quad < \quad \left(z_0 + \Phi\right) \exp\left(-|c_2| \frac{\sum x_i y_i}{|z_0| \sum x_i^2}\right) - \Phi. \tag{58}$$

*Proof.* Note that the quantity $z_k + \Phi$ is shrinking exponentially as $\dot{X}_t = -2\eta|z_k|X_t$ and $w_k$ is shrinking exponentially as $\dot{X}_t = -\eta^2 z_k^2 X_t$. We have that $\eta < \frac{2}{|z_0|} < \frac{2}{|z_k|}$, thus the rate at which $z_k + \Phi$ is decreasing is faster than $w_k$, this implies that at a certain point we will have $w_k > -2z_k - 2\Phi$. Let us study this time $\tau_2$. Note that the fact that $|z_0| > |z_k| > \Phi$ implies that $2\eta|z_k| > 2\eta\Phi$ and $\eta^2|z_k| > \eta^2\left(\Phi\right)^2$

$$2\eta|z_0| \quad > \quad 2\eta|z_k| \quad > \quad 2\frac{\sum x_i y_i}{|z_0| \sum x_i^2} \tag{59}$$

and

$$\eta^2|z_0|^2 \quad > \quad \eta^2|z_k|^2 \quad > \quad \frac{\left(\Phi\right)^2}{z_0^2}. \tag{60}$$

By the time $\tau_2 = \min_k \left\{w_0 \geq -2z_0 - 2\Phi, \; z_k > -\Phi\right\}$ we have the thesis. Note that definition of $\tau_2$ and the iterative formulas

$$w_{\tau_1} = w_0 \prod_{k=1}^{\tau_2}(1 - \eta^2 z_k^2), \quad and$$

$$z_{\tau_1} + \Phi = (z_0 + \Phi) \prod_{k=1}^{\tau_2}(1 - 2\eta z_k) \tag{61}$$

imply that

$$\prod_{k=0}^{\tau_2-1}(1 - \eta^2 z_k^2)\left|\frac{w_0}{z_0 + \Phi}\right| = \left|\frac{w_{\tau_2}}{z_0 + \Phi}\right| > 2\frac{z_{\tau_2} + \Phi}{z_0 + \Phi} = 2\prod_{k=0}^{\tau_2-1}(1 - 2\eta z_k). \tag{62}$$

This implies

$$\left|\frac{w_0}{z_0 + \Phi}\right| > 2\prod_{k=0}^{\tau_2-1}\frac{1 - 2\eta z_k}{1 - \eta^2 z_k^2} > 2\left(\frac{1 - 2\eta z_0}{1 - \eta^2\left(\Phi\right)^2}\right)^{\tau_2} > 2\left(1 - 2\eta z_0\right)^{\tau_2}. \tag{63}$$

Analogously we have

$$\left|\frac{w_0}{z_0 + \Phi}\right| < 2\prod_{k=0}^{\tau_2-2}\frac{1 - 2\eta z_k}{1 - \eta^2 z_k^2} = 2\prod_{k=0}^{\tau_2-2}\frac{1 - 4\eta^2 z_k^2}{1 - \eta^2 z_k^2}\frac{1}{1 + 2\eta|z_k|} < 2\prod_{k=0}^{\tau_2-2}\left(1 + 2\eta\Phi\right)^{-1}. \tag{64}$$

Thus defining $c_2 := \log\left(\frac{1}{2}\left|\frac{w_0}{z_0 + \Phi}\right|\right) < 0$ we obtain

$$\begin{aligned} -c_2 &< \quad 2\eta|z_0|\tau_2 \\ -c_2 &> \quad \log\left(1 + 2\eta\Phi\right)\left(\tau_2 - 1\right). \end{aligned} \tag{65}$$

Thus we conclude that

$$
\frac{|c_2|}{2\eta|z_0|} \quad < \quad \tau_2
$$

$$
\frac{|c_2|}{2\eta\Phi}\left(1 - \eta\Phi\right)^{-1} + 1 \quad > \quad \frac{|c_2|}{\log\left(1 + 2\eta\Phi\right)} + 1 \quad > \quad \tau_2
$$

(66)

This implies that

$$
(z_0 + \Phi)\exp\left(-|c_2|\frac{\sum x_i y_i}{|z_0|\sum x_i^2}\right) - \Phi \quad > \quad z_{\tau_2}
$$

(67)

and

$$
(z_0 + \Phi)\left(1 - 2\eta|z_0|\right)\exp\left(-|c_2|\frac{|z_0|\sum x_i^2}{\sum x_i y_i}\left(1 - \eta\Phi\right)^{-1}\right) - \Phi \quad < \quad z_{\tau_2}.
$$

(68)

Analogously

$$
w_0(1 - \eta^2 z_0^2)^{\frac{|c_2|}{\log(1+2\eta\Phi)}+1} \quad < \quad w_{\tau_2} \quad < \quad w_0\left(1 - \eta^2\left(\Phi\right)^2\right)^{\frac{|c_2|}{2\eta|z_0|}}.
$$

(69)

This means approximately that

$$
w_0(1 - \eta^2 z_0^2)\exp\left(-\frac{\eta|c_2|}{2}\frac{z_0^2\sum x_i^2}{\sum x_i y_i}\right) \quad < \quad w_{\tau_2} \quad < \quad w_0\exp\left(-\frac{\eta|c_2|}{2}\frac{[\sum x_i y_i]^2}{z_0\left[\sum x_i^2\right]^2}\right).
$$

(70)

$\square$

**Lemma 16** (Rate of convergence 2.). *Let* $c_2 := \left|\sum_i|\mathbf{a}_i^2(0) - \mathbf{b}_i^2(0)|/\mathbf{a}(0)^\top\mathbf{b}(0)\right|$. *If* $0 < \sum_i|\mathbf{a}_i^2(0) - \mathbf{b}_i^2(0)| \leq -2\mathbf{a}(0)^\top\mathbf{b}(0)$, *then by time* $\tau_2$ *as below we are under the assumptions of Lemma 13, so we have* $\sum_i|a_i^2(\tau_2) - b_i^2(\tau_2)| \geq -2a(\tau_2)^\top b(\tau_2) > 0$ *and we have*

- $\frac{|c_2|}{2\eta|\mathbf{a}(0)^\top\mathbf{b}(0)-\Phi|} \quad < \quad \tau_2,$

- $\tau_2 \quad < \quad \frac{|c_2|}{\log(1+2\eta\Phi)} + 1,$

- $\sum_i|\mathbf{a}_i^2(0) - \mathbf{b}_i^2(0)|(1 - \eta^2 z_0^2)\exp\left(-\frac{\eta|c_2|}{2}\frac{z_0^2\sum x_i^2}{\sum x_i y_i}\right) \quad < \quad \sum_i|a_i^2(\tau_2) - b_i^2(\tau_2)|,$

- $\sum_i|a_i^2(\tau_2) - b_i^2(\tau_2)| \quad < \quad \sum_i|\mathbf{a}_i^2(0) - \mathbf{b}_i^2(0)|\exp\left(-\frac{\eta|c_2|}{2}\frac{[\sum x_i y_i]^2}{z_0\left[\sum x_i^2\right]^2}\right),$

- $(z_0 + \Phi)\left(1 - 2\eta|z_0|\right)\exp\left(-|c_2|\frac{|z_0|\sum x_i^2}{\sum x_i y_i}\left(1 - \eta\Phi\right)^{-1}\right) \quad < \quad a(\tau_2)^\top b(\tau_2),$ *and*

- $a(\tau_2)^\top b(\tau_2) \quad < \quad (z_0 + \Phi)\exp\left(-|c_2|\frac{\sum x_i y_i}{|z_0|\sum x_i^2}\right).$

*Proof.* One bound comes from Lemma 15 and Lemma 11. The other one comes by just following the proof of Lemma 15. $\square$

This concludes the proof of Proposition 4 and shows that the $Q_i$ are lower bounded for all $i$ when initialization is in Area C and $\eta < \min\left\{\frac{1}{2|\varepsilon|}, \frac{2}{\bar{\lambda}}\right\}$.

# F   CONVERGENCE SPEED CASE BY CASE

This section serves as merger for all the theory made before. Precisely, here we use the analysis developed to prove Theorem 2 and Proposition 1.

We prove below and in Appendix E that in the three different regions of the landscape we have different PL constants $\mu$ for $\varepsilon$-and then for $L$. Precisely, if $\varepsilon \geq 0$ then $\mu > 2\Phi$, if $\varepsilon < -\Phi/2$ then $\mu = Q(\tau)$, and if $-\Phi/2 \leq \varepsilon < 0$ then $\mu = \Phi$. This implies that we have convergence with the minimum of $Q(\tau)$ and $2\Phi$ as PL constant until $|\varepsilon| > \Phi/2$, then we have convergence with $\Phi$ as PL constant from then on.

### F.1 Positive residuals

First note that $\eta \boldsymbol{\lambda} \varepsilon > \eta^2 \varepsilon^2 (\varepsilon + \Phi)$, indeed $\boldsymbol{\lambda} > 2(\varepsilon + \Phi)$ by Cauchy Schwartz and $\eta \leq \frac{\sqrt{2}}{\varepsilon} < \frac{2}{\varepsilon}$. This implies that when $\eta$ is infinitesimal, the gain is at least

$$
\begin{aligned}
\varepsilon(k+1) &= (1 - \eta \boldsymbol{\lambda})\varepsilon(k) + \eta^2 \varepsilon(\varepsilon + \Phi) \\
&\leq \varepsilon(k) - \eta \boldsymbol{\lambda} \varepsilon(k)\left(1 - \frac{\eta}{2}\varepsilon(k)\right) \\
&\leq \left(1 - \frac{2 - \sqrt{2}}{2}\eta \boldsymbol{\lambda}\right)\varepsilon(k).
\end{aligned}
\tag{71}
$$

Next note that $\frac{x}{\sqrt{x^2 + y^2}} = \sqrt{1 - \frac{y^2}{x^2 + y^2}} \leq 1 - \frac{1}{2}\frac{y^2}{x^2 + y^2}$. When $\eta \sim \frac{2}{\boldsymbol{\lambda}}(1 - \delta), \delta > 0$ we have that

$$
\begin{aligned}
|\varepsilon(k+1)| &= \left|(1 - \eta \boldsymbol{\lambda})\varepsilon(k) + \eta^2 \varepsilon^2(k)(\varepsilon(k) + \Phi)\right| \\
&\leq \left|\varepsilon(k) - \frac{2\boldsymbol{\lambda}(1 - \delta)}{\sqrt{\boldsymbol{\lambda}^2 + 4\Phi^2}}\varepsilon(k) + \frac{4(1 - \delta)^2}{\boldsymbol{\lambda}^2 + 4\Phi^2}\varepsilon(k)^2(\varepsilon(k) + \Phi)\right| \\
&\leq \left|(-1 + 2\delta)\varepsilon(k) + \frac{4\Phi^2(1 - \delta)}{\boldsymbol{\lambda}^2 + 4\Phi^2}\varepsilon(k) + \frac{4(1 - \delta)}{\boldsymbol{\lambda}^2 + 4\Phi^2}\varepsilon(k)^2(\varepsilon(k) + \Phi)\right| \\
&\leq (1 - 2\delta)\varepsilon(k).
\end{aligned}
\tag{72}
$$

This implies that within our learning rate boundaries we have exponential convergence with rate either controlled by $\eta$ or $\delta$ at power 1.

In case $\frac{2}{\boldsymbol{\lambda}} \leq \eta \leq \frac{2}{\boldsymbol{\lambda}}(1 - \delta)$ then convergence happens exponentially but in time $O(\eta^{-2})$. For instance $\eta \sim \frac{2}{\boldsymbol{\lambda}}$ we have that

$$
\begin{aligned}
|\varepsilon(k+1)| &= \left|(1 - \eta \boldsymbol{\lambda})\varepsilon(k) + \eta^2 \varepsilon^2(k)(\varepsilon(k) + \Phi)\right| \\
&\leq \left|\varepsilon(k) - \frac{2\boldsymbol{\lambda}}{\sqrt{\boldsymbol{\lambda}^2 + 4\Phi^2}}\varepsilon(k) + \frac{4}{\boldsymbol{\lambda}^2 + 4\Phi^2}\varepsilon(k)^2(\varepsilon(k) + \Phi)\right| \\
&\leq \left|-\varepsilon(k) + \frac{4\Phi^2}{\boldsymbol{\lambda}^2 + 4\Phi^2}\varepsilon(k) + \frac{4}{\boldsymbol{\lambda}^2 + 4\Phi^2}\varepsilon(k)^2(\varepsilon(k) + \Phi)\right| \\
&\leq (1 - \eta^2\Phi^2)\varepsilon(k).
\end{aligned}
\tag{73}
$$

In case $\eta \geq \frac{2}{\boldsymbol{\lambda}}$ but convergence happen, then convergence happens only at most logarithmically fast at least for a first phase, precisely

$$
\begin{aligned}
|\varepsilon(k+1)| &= \left|(1 - \eta \boldsymbol{\lambda})\varepsilon(k) + \eta^2 \varepsilon^2(k)(\varepsilon(k) + \Phi)\right| \\
&\leq \left|-\varepsilon(k) + \frac{4}{\boldsymbol{\lambda}^2}\varepsilon(k)^2(\varepsilon(k) + \Phi)\right| \\
&\leq \left(1 - \frac{4}{\boldsymbol{\lambda}^2}\Phi\varepsilon(k)\right)\varepsilon(k).
\end{aligned}
\tag{74}
$$

### F.2 Negative residuals $-\Phi/2 < \varepsilon < 0$

When the residuals are small negative we have exponential convergence, precisely, for very small $\eta \ll 1$ we have rate at least $(1 - \eta\Phi)$:

$$
\begin{aligned}
|\varepsilon(k+1)| &= \left|(1 - \eta \boldsymbol{\lambda})\varepsilon(k) + \eta^2 \varepsilon^2(k)(\varepsilon(k) + \Phi)\right| \\
&\leq \left|(1 - \eta \boldsymbol{\lambda}) + \frac{\eta^2}{4}\Phi^2\right||\varepsilon(k)| \\
&\leq (1 - \eta\Phi)|\varepsilon(k)|.
\end{aligned}
\tag{75}
$$

For bigger $\eta = \frac{2}{\lambda}$, we have convergence with rate about $\sim 2$. The maximum over $\boldsymbol{\lambda}(0), \varepsilon(k)$ in the region in which $\varepsilon(k) = -c\frac{\Phi}{2}$ with $c \in (0, 1]$

$$\max |\varepsilon(k+1)| = \max |(1 - \eta\boldsymbol{\lambda})\varepsilon(k) + \eta^2\varepsilon^2(k)(\varepsilon(k) + \Phi)|$$

$$\leq \max \left| \varepsilon(k) - \frac{2\boldsymbol{\lambda}}{\sqrt{\boldsymbol{\lambda}(0)^2 + 4\Phi^2}}\varepsilon(k) + \frac{4}{\boldsymbol{\lambda}(0)^2 + 4\Phi^2}\varepsilon(k)^2(\varepsilon(k) + \Phi) \right|. \tag{76}$$

Note that the minimum in $\boldsymbol{\lambda}(0)$ of this last equation is for $\sqrt{\boldsymbol{\lambda}(0) + 4\Phi^2} = \delta + 2\Phi$ for some $\delta > 0$ which satisfies $\delta \ll 1$. This is independent of the size of $\varepsilon$. Along this trajectory, $\mathbf{a}^\top\mathbf{b} = (1 - c/2)\Phi$ and $\boldsymbol{\lambda} \leq (2 - c)\Phi + \delta$ This implies

$$\max_{\boldsymbol{\lambda}(0),\varepsilon(k)} |\varepsilon(k+1)| \leq \max_{\varepsilon(k)} \left| \varepsilon(k) - \frac{2(2-c)\Phi}{\delta + 2\Phi}\varepsilon(k) + \frac{4}{(\delta + 2\Phi)^2}\varepsilon(k)^2(\varepsilon(k) + \Phi) \right|$$

$$\leq \max_{\varepsilon(k)} \left| -\frac{(2 - 2c)\Phi + \delta}{\delta + 2\Phi} + \frac{c(2 - c)}{(\delta + 2\Phi)^2}\Phi^2 \right| |\varepsilon(k)|$$

$$\leq \left| -1 + c\frac{2\Phi}{\delta + 2\Phi} + c(2 - c)\frac{\Phi^2}{(\delta + 2\Phi)^2} \right| |\varepsilon(k)| \leq \left| c - 1 - \frac{c^2}{4} + \frac{c}{2} \right| |\varepsilon(k)|$$

$$\underset{c=1}{\leq} \frac{1}{4}|\varepsilon(k)| = \frac{1}{8}\Phi. \tag{77}$$

The maximum of $|c^2/4 - 3c/2 + 1|$ over $c \in (0, 1]$ is $c = 1$.

In the case of $c = 1$, on the next step, in this case, we are in the positive residuals setting with $\boldsymbol{\lambda}$ as follows $2 \cdot \mathbf{a}^\top\mathbf{b} = \frac{9}{4}\Phi + \delta$. Here, then

$$|\varepsilon(k+2)| \leq \left| -\frac{5}{4} + \frac{1}{4\Phi^2}\varepsilon(k+1)(\varepsilon(k+1) + \Phi) \right| \varepsilon(k+1)$$

$$\leq \frac{1}{16}\left( 5 - \frac{1}{16} \right)|\varepsilon(k)|. \tag{78}$$

So after 2 steps, we had a linear shrink of $5/16$ and the linear convergence with constant $\mu = \Phi$ restarts, this is the plus 2 of the theorem.

### F.3 NEGATIVE RESIDUALS $\varepsilon \leq \Phi/2$

This case is taken care of in Appendix E until $\varepsilon = 0$. With the same $\mu > 0$ we have exponential convergence until $\Phi/2$. As we said in Appendix E as $\varepsilon$ crosses $\Phi$, the norm $\boldsymbol{\lambda}$ restarts increasing. This implies that a good lower bound remains $Q$ of the time of crossing. The evolution of $\varepsilon$

$$\varepsilon(k+1) = (1 - \eta\boldsymbol{\lambda})\varepsilon(k) + \eta^2\varepsilon(k)^2(\varepsilon(k) + \Phi) \geq (1 - \eta Q)\varepsilon(k). \tag{79}$$

The time $t$ taken to $\varepsilon$ to go from $\Phi$ to $\Phi/2$ is thus

$$\Phi/2 \geq (1 - \eta Q)^t\Phi \tag{80}$$

so we have

$$t \leq \frac{\log(\Phi) - \log(\Phi/2)}{-\log(1 - \eta Q)} \leq \frac{\log(\Phi) - \log(\Phi/2)}{\eta Q_\tau}. \tag{81}$$

### F.4 CLOSING UP: TIGHT RATE

The previous sections and Lemma 5 allow us to conclude that we have loss convergence, i.e., $L \leq \delta$, in a number of steps which is

$$t \leq \tau + \frac{\log(\Phi) - \log(\Phi/2)}{\eta \min\{Q_\tau, 2\Phi\}} + 2 + \frac{\log(\Phi/2) - \log(\delta)}{\eta \min\{Q_\tau, \Phi\}}, \tag{82}$$

where $\tau$ is the $\tau_1$ defined in Definition 4 and evaluated in Proposition 4. This establishes Theorem 2.

# G  CURIOSITY: JUMPS BETWEEN REGIONS

Note that if the dynamics does not jump from one side to the other of the landscape, then we have a clean exponential convergence and we can control the implicit regularization. We will see under which hypothesis on the learning rate this happens.

Note that Equation 5 tells us that after one step $\varepsilon$ does not change sign (thus you remain in the same region in which you started) if and only if we have the following bound on the learning rate.

**Definition 5.** For all $\mathbf{a}, \mathbf{b} \in \mathbb{R}^n$, let $\alpha = \frac{\varepsilon(\varepsilon+\Phi)}{\lambda^3}$, define

$$\eta_1 \quad := \quad \frac{1}{\lambda}\left(1 + \alpha + 2\alpha^2 + 5\alpha^3 + 14\alpha^8 + \ldots\right), \tag{83}$$

$$\eta_2 \quad := \quad \frac{2}{\lambda}\left(1 + 2\alpha + 8\alpha^2 + 40\alpha^3 + 224\alpha^4 + \ldots\right). \tag{84}$$

The way we obtain $\eta_1$ is by seeing for what $\eta$ we have that $\varepsilon(t+1) = 0$. Precisely,

**Lemma 17.** *If $\eta = \eta_1$, we have that the residuals at the next steps are 0. If $\eta = \eta_2$, then the residuals at the next steps are the same but changed of sign. Moreover,*

- *If $\eta \in (0, \eta_1)$ then $sign(\varepsilon(1)) = sign(\varepsilon)$ and $|\varepsilon(1)| < |\varepsilon|$.*

- *If $\eta \in (\eta_1, \eta_2)$ then $sign(\varepsilon(1)) \neq sign(\varepsilon)$ and $|\varepsilon(1)| < |\varepsilon|$.*

*Proof of Lemma 17.* Note that the residuals after one step are the same sign as the previous residuals if and only if

$$1 - \eta\lambda + \eta^2\varepsilon(\varepsilon + \Phi) \quad \geq \quad 0. \tag{85}$$

Solving this one as a second degree equation gives

$$\eta \quad \leq \quad \frac{\lambda - \sqrt{\lambda^2 - 4\varepsilon(\varepsilon+\Phi)}}{2\varepsilon(\varepsilon+\Phi)} \quad \text{or} \quad \eta \quad \geq \quad \frac{\lambda + \sqrt{\lambda^2 - 4\varepsilon(\varepsilon+\Phi)}}{2\varepsilon(\varepsilon+\Phi)} \tag{86}$$

Now expanding in Taylor the square root, we obtain that

$$\eta_1 \quad \leq \quad \frac{1}{2\varepsilon(\varepsilon+\Phi)}\left(\frac{4\varepsilon(\varepsilon+\Phi)}{2\lambda} + \frac{16\varepsilon^2(\varepsilon+\Phi)^2}{8\lambda^3} + \ldots\right) \quad = \quad \frac{1}{\lambda} + \frac{\varepsilon(\varepsilon+\Phi)}{\lambda^3} + \ldots \tag{87}$$

This implies that the residuals are the same sign as the starting ones if

$$\eta \quad \leq \quad \eta_1 \quad \text{or} \quad \eta \quad \geq \quad \frac{\lambda}{\varepsilon(\varepsilon+\Phi)} - \eta_1. \tag{88}$$

Analogously, for $\eta_2$ we have that the absolute value of the residuals is smaller than the absolute value of the residuals one step before, if and only if

$$2 - \eta\lambda + \eta^2\varepsilon(\varepsilon + \Phi) \quad \geq \quad 0. \tag{89}$$

This implies that

$$\eta \quad \leq \quad \frac{\lambda - \sqrt{\lambda^2 - 8\varepsilon(\varepsilon+\Phi)}}{2\varepsilon(\varepsilon+\Phi)} \quad \text{or} \quad \eta \quad \geq \quad \frac{\lambda + \sqrt{\lambda^2 - 8\varepsilon(\varepsilon+\Phi)}}{2\varepsilon(\varepsilon+\Phi)} \tag{90}$$

and analogously to before

$$\eta \quad \leq \quad \eta_2 \quad \text{or} \quad \eta \quad \geq \quad \frac{\lambda}{\varepsilon(\varepsilon+\Phi)} - \eta_2. \tag{91}$$

Also note that for $\varepsilon > 0$ we have that $\varepsilon(1) < \varepsilon$ or for $\varepsilon < 0$ we have that $\varepsilon(1) > \varepsilon$ if and only if

$$1 - \eta\lambda + \eta^2\varepsilon(\varepsilon + \Phi) \quad \leq \quad 1. \tag{92}$$

This solves when

$$\eta \quad \leq \quad \frac{\lambda}{\varepsilon(\varepsilon + \Phi)} \tag{93}$$

$\square$

Note that what we did here implies that if $\eta \leq \eta_2$ and $\eta \leq \frac{\sqrt{2}}{\varepsilon}$ for all the $\boldsymbol{\lambda}$s along the trajectory we thus always have exponential convergence if such PL condition holds. We know from the previous section that in this setting $\boldsymbol{\lambda}$ is always smaller than $\bar{\boldsymbol{\lambda}}$. So if such a $\mu$ exists and $\eta \leq \eta_2$ with $\bar{\boldsymbol{\lambda}}$ and we converge and we can properly bound the implicit regularization.

# H LOCATION OF CONVERGENCE - PROOF OF THEOREM 1

We will bound here the final $Q$ for two reasons:

- Understanding the location of convergence.
- Picking the right learning rate.

Note that assuming $\eta \leq \frac{\sqrt{2}}{\varepsilon}$ along the whole trajectory we have that $Q$ strictly monotonically shrinks along the trajectory. This means that the dynamics may seem to oscillate around in an uncontrollable way, but every time it oscillates is landing on a trajectory that takes to a global minimum with lower $Q$.

Note that this is true almost everywhere, indeed if the trajectory is such that at a certain point in time $t$ satisfies $\eta = \varepsilon(t)^{-1}$ exactly, then the trajectory would land on the trajectory taking to the saddle, indeed

$$\mathbf{a}(t+1) = -\mathbf{b}(t+1) = \mathbf{a}(t) - \mathbf{b}(t). \tag{94}$$

Luckily, fixing a learning rate size, the set of starting points for which this is the case has measure zero. Observe also that $\eta = -\varepsilon^{-1}$ is instead optimal and results in $\mathbf{a}(t+1) = \mathbf{b}(t+1)$, implying convergence to a balanced solution. This means that assuming $\eta \leq \frac{\sqrt{2}}{\varepsilon}$ implies that the dynamics may diverge or converge, but for sure at every step is getting closer and closer to the subspace in which $\mathbf{a} = \mathbf{b}$. Moreover, note that all the $Q_i$ change sign if and only if $\eta|\varepsilon| > 1$.

Regarding the proof of the upperbound of Theorem 1 note that for all $t$

$$Q_i(t) \quad = \quad Q_i(0) \cdot \prod_{k=0}^{t-1}(1 - \eta^2 \varepsilon_k^2) \quad = \quad Q_i(0) \cdot \exp\left(\sum_{k=0}^{t-1} \log(1 - \eta^2 \varepsilon_k^2)\right). \tag{95}$$

In absolute value, we can thus upperbound the RHS as follows, by applying the Taylor expansion whenever $\eta|\varepsilon| < 1$

**Lemma 18** (Upperbound to the inbalance, 1). *Let $\eta|\varepsilon(t)| < 1$ for all $t \in \mathbb{N}$, then for all $t \in \mathbb{N}$ we have*

$$|Q_i(t)| \quad < \quad |Q_i(0)| \cdot \exp\left(-\eta^2 \sum_{k=0}^{t-1} \varepsilon_k^2\right).$$

By combining this lemma and Lemma **??** we obtain

**Lemma 19** (Upperbound to the inbalance, 2). *Let $\eta|\varepsilon(0)| < 1$ and $\eta \leq \tilde{\eta}$, then for all $t \in \mathbb{N}$*

$$|Q_i(t| \quad < \quad |Q_i(0)| \cdot \exp\left(-\eta^2 \sum_{k=0}^{t-1} \varepsilon_k^2\right).$$

This establishes the upper bound of Theorem 1. Regarding the proof of the lower bound, notice that we have from Appendix D.3 that the rate of convergence of $\varepsilon$ is at least $Q(\tau_1)$ in Area B and at least $2\Phi$ in Area A, once adding the right assumption on the learning rate. This implies that if the initialization is in Area B or C, then

**Lemma 20** (Lower bound to the inbalance). *Assume there exists $\tilde{t}$ such that for all $t \geq \tilde{t}$ we have $\eta|\varepsilon(0)| < 1/2$ then*

$$Q_i(t)$$

*Proof.* Note that the fact that $\eta|\varepsilon(k)| < 1/2$ for all $k$ makes sure that

$$Q_i(t) = Q_i(0) \cdot \prod_{k=0}^{t-1}(1 - \eta^2 \varepsilon_k^2) \geq Q_i(0) \cdot \exp\left(-\sum_{k=0}^{t-1} \eta^2 \varepsilon_k^2 - \eta^4 \varepsilon_k^4\right) \tag{96}$$

since for $x \in [0, 1/2]$ we have $1 - x > e^{-x - x^2}$. Next note that in these hypothesis, by Theorem 2, we ahve exponential convergence, thus

$$
\begin{aligned}
Q_i(t) \;\geq\; & Q(0) \cdot \exp\left(-\eta^2 \varepsilon(0)^2 \sum_{k=1}^{\infty}(1 - \eta Q(\tau))^{2k} - \eta^4 \varepsilon_0^4 \sum_{1}^{\infty}(1 - \eta Q(\tau))^{4k}\right) \\
\geq\; & Q(0) \cdot \exp\left(-\frac{\eta \varepsilon(0)^2}{Q(\tau)(2 - \eta Q(\tau))} - \frac{\eta^3 \varepsilon(0)^4}{Q(\tau)(8 - \eta Q(\tau))}\right) \\
\geq\; & Q(0) \cdot \exp\left(-\frac{\sqrt{\eta}\,\varepsilon(0)^2}{2\Phi}\left(1 + \frac{\eta^2 \varepsilon(0)^2}{8}\right)\right) \\
\geq\; & Q(0) \cdot \exp\left(-\frac{\sqrt{\eta}\,\varepsilon(0)^2}{\Phi}\right).
\end{aligned}
\tag{97}
$$

By plugging in the lower bound in Lemma 14. This concludes the proof of the lemma. $\qquad\square$

This concludes the proof of Theorem 1.

