# OpenReview forum: "Gradient Descent Converges Linearly to Flatter Minima than Gradient Flow in Shallow Linear Networks"
_ICLR.cc/2025/Conference — Submitted to ICLR 2025_

### Official Review · Reviewer_WY3N · 2024-10-29

**Soundness:** 1
**Presentation:** 1
**Contribution:** 1
**Rating:** 3
**Confidence:** 4

**Summary:**

The paper analyzes the GD dynamic of a 2-layer linear network with scalar input and scalar output. The authors study the rate of convergence of GD and provide some properties of the reached solution by using conserved quantities during a gradient flow dynamic.

**Strengths:**

The topic is relevant and the approach of studying simplified models is worthwhile.
The paper includes clear and well-designed figures.

**Weaknesses:**

General impression: The paper appears to have been written hastily, lacking structure, with numerous typographical errors, missing hypotheses, and unsatisfactory proofs that are difficult to follow. Major revisions requiring another round of review are necessary in my opinion.

Regarding the results: The contributions seem overall quite weak and insufficient for acceptance at ICLR.

Some comments/suggestions:

1. The authors state
> Despite its simplicity, the objective (2), which has also been studied by prior work (Lewkowycz et al., 2020; Wang et al., 2022; Chen & Bruna, 2023; Ahn et al., 2024), is a useful object of study because...

By quickly checking some of these references, it appears they do not restrict their analysis to the case $x_i, y_i \in \mathbb{R}$ only.

2. The authors claim
> In addition to showing how fast gradient descent converges to some global minimizer, we can also describe which of the many possible solutions, a ⊤b = Φ, gradient descent will converge to (l. 083)

The corresponding result in section 3 does not specify which solution is reached, but only describes certain properties of that minimizer (which could be satisfied by multiple solutions).

3. Furthermore, the analysis for Theorem 1 requires $Q(0) \neq 0$ at initialization. This should be stated in the hypotheses of Theorem 1 and in the paragraph following equation (7). Additionally, $Q(0) = 0$ is also a common hypothesis, this should be discussed.

4. The paragraph following equation (8) motivates Takeaway 1 but falls short of rigorously proving it.

5. Sections 4, 5, and 6 severely lack structure (Section 5, subsection 5.1??).

6. Appendix C is unclear:
- First, the conserved quantity is preserved for gradient flow (not necessarily gradient descent). Therefore, the conclusion
> These two lemmas imply that the norm of the solution found by gradient descent will always be smaller than λ∞ and since ε∞ = 0 we can compute it using the formula above

(note: two → one lemma?) about a solution found by *gradient descent* requires proof.
- There appears to be confusion between $\lambda$ before Lemma 7 (continuous) and after (discrete); the link is not examined.
- The computations for the "proof" of Lemma 7 are incorrect. While the result may be true, a proper proof is needed.
- not clear at what point is defined the maximal sharpness (the formulation of Lemma 7 suggests it is at initialization).
- For Lemma 8, $\eta$ needs to be less than 2/maximal sharpness; this should likely be also the case in Lemma 7 (the first inequality of Lemma 7 appears to lack any attempt of proof)?

**Questions:**

See above

and l. 068

> More importantly, the dynamics when optimizing (2) with gradient descent are qualitatively similar to the dynamics of training more complex models

Could the authors elaborate on this point?

---

> ### Author Response · Authors · 2024-11-21
> **Response to WY3N**
>
> Although it is always difficult to find a negative review, we appreciate and thank a lot Reviewer WY3N for the extremely thorough and detailed comments. They were all very helpful and on point, and they helped us reshape the paper substantially. Below a detailed answer to every comment:
>
> *Regarding your general impression:*
> Thanks for the comment, major revision have been made, as you can notice from the new pdf uploaded. We have clarified a lot of things, and we pointed to the proofs which have been now partly reorganized. We hope that these revisions make the paper clearer.
>
> *Regarding the results:*
> We disagree on the insufficiency of the contributions, although this is somewhat personal. We better clarified the implications of the results in the conclusions if that could be of better help. Moreover, many related papers have been published at Neurips, ICLR, ICML so seem to be interesting to the community.
>
> *1. About related work:* It is true that the model we consider is quite simple, and has already been studied in various forms in the existing literature. That said, we believe that we have several significant contributions to make:
> - our analysis is more detailed than that of previous work (see discussion in section 1),
> - this additional detail allows us to prove a guarantee about the rate of convergence which is lacking in related work,
> - the detail also allows us to closely characterize the degree of implicit regularization caused by gradient descent, and
> - putting these together we are able to highlight an interesting phenomenon where convergence speed and implicit regularization apparently trade off against each other.
>
> While other papers may study the same model (or more complex/``realistic'' models similar to ours), none of them include analysis at this level of detail, and therefore they do not arrive at our conclusions, which we think are interesting and could form the basis for additional study. Certainly, we would like to extend our analysis to more complex models, but this presents a significant technical challenge that must be left for future work.
>
> *2.:* Thanks for pointing out, we changed this claim.
>
> *3. Regarding $Q$:* Actually not, the analysis works perfectly also with $Q=0$ (in this case, it stays zero forever), we slightly changed the statement to clarify. Thanks a lot for pointing out.
>
> *4. Takeaway 1:* Takeaway 1 is a direct implication of Theorem 1. We reorganized and reworded the section to make it clear. Thanks for pointing out the confusion.
>
> *5. Regarding Sections 5-7:* We believe that the main point of confusion in the presentation, as raised by all the reviewer, comes from the fact that section 6 had to be subsection 5.2. Unfortunately, we accidentally made a subsection into a section. This has been solved right now and the section has been clarified further. Moreover, we updated the lemmas to slightly stronger versions proved in the newly designed Appendix C.
> Contentwise, we made clear that Section 5 deals with how the regimes of optimization change given the size of the learning rate and Section 6 regards a notion we introduced to prove linear convergence.
>
> *6. We completely reframed Appendix C:* Thanks a lot for pointing out all this. We added lemmas and proofs, now not only it is more clear, but the missing proofs are present and the results for gradient descent are actually stronger. See also Section 5.1 to check the improvements.
>
> *Regarding similarity in dynamics:*
> Our model is simple, so it certainly will not capture all of the relevant properties of realistic neural networks, but we believe it does capture some important ones. As an example, a lot of recent work (e.g. [Cohen et al 2021]) has studied the Edge of Stability (EoS) phenomenon and it appears that training hyperparameters are typically selected in such a way that neural networks are often trained in the EoS regime. Nobody knows exactly how typical overparametrized neural networks are able to generalize as well as they do, but implicit regularization is theorized to play a role, and our work suggests that the implicit regularization may be enhanced by training in the EoS regime, which might partially explain why people train models in that way. There is also experimental evidence in some of the cited related work including a newly-added citation (see Figure 1 in [Xu and Ziyin 2024]) that relates certain properties of training models like ours with training more complex and realistic models.

---

> ### Author Response · Authors · 2024-11-27
>
> Dear reviewer WY3N,
>
> We value a lot your advice and opinions.
> Our deadline for submitting the revised manuscript is tonight. If possible, could you please share any additional comments or suggestions before then? This would allow us to incorporate your feedback in the rebuttal manuscript!
>
> Thank you very much again for your time!
>
> Best,
>
> The Authors

---

### Official Review · Reviewer_ZTfs · 2024-11-04

**Soundness:** 4
**Presentation:** 3
**Contribution:** 2
**Rating:** 6
**Confidence:** 3

**Summary:**

The paper aims to study the GD dynamics of two-layer linear networks with scalar input and output. The authors present a detailed analysis of the model they introduce, being able to identify analytically all the relevant aspects of the setting. In particular, they show the ability of GD to converge for unexpectedly large learning rates (edge of stability), and a full characterization of the implicit regularization that GD brings. Finally, they merge these two aspects and show a trade-off between speed and regularization.

**Strengths:**

- All results presented are precise and analytical, presented with mathematical rigor and well written. The model is extensively analyzed and all possible aspects are discussed in detail.
- I think the most valuable takeaway from this paper is the proof that, even in a very simple model, gradient flow dynamics are inherently different from gradient descent. In particular, the paper shows that GD regularizes better in this setting, but more generally can be used as a proof that the common use of GF as a theoretical tool for understanding GD is not always well founded.

**Weaknesses:**

- I think the claim of studying flat linear networks is an overstatement. The setting is too simple to claim that it is a faithful model for networks. It seems to me that related work analyzing similar settings is either not as simplified as this, or is motivated by something else like a matrix factorization problem (or both).
- Alongside the previous point, I am not sure that the results presented are relevant enough to pass the bar of the conference.

**Questions:**

- Do the authors have any idea how to extend some of the exposed results to non-scalar inputs? For example, instead of $\vec x\in \mathbb{R}$, at least $\vec x\in \mathbb{R^2}$ ?
- Section 5 is less than a quarter page long. Ether there is a problem/ something missing, or it should be merged together with Section 6 and better contextualized.
Minor:
- I think displaying $\lambda$ in bold can be misleading, since in the rest of the paper the bold is used for vectors.
- Typo line 430: $\eta<\eta$.

---

> ### Author Response · Authors · 2024-11-21
> **Response to ZTfs**
>
> We want to thank Reviewer ZTfs for the comments and praises.
>
> *Regarding the significance and relationship to prior work:*
>
> It is true that the model we consider is quite simple, and has already been studied in various forms in the existing literature. That said, we believe that we have several significant contributions to make:
> - our analysis is more detailed than that of previous work (see discussion in section 1),
> - this additional detail allows us to prove a guarantee about the rate of convergence which is lacking in related work,
> - the detail also allows us to closely characterize the degree of implicit regularization caused by gradient descent, and
> - putting these together we are able to highlight an interesting phenomenon where convergence speed and implicit regularization apparently trade off against each other.
>
> While other papers may study the same model (or more complex/``realistic'' models similar to ours), none of them include analysis at this level of detail, and therefore they do not arrive at our conclusions, which we think are interesting and could form the basis for additional study. Certainly, we would like to extend our analysis to more complex models, but this presents a significant technical challenge that must be left for future work.
>
> *Regarding the relevance of the model:*
> Although our model is simple, and it certainly will not capture all of the relevant properties of realistic neural networks, we believe it does capture some of them. As an example, a lot of recent work (e.g. [Cohen et al 2021]) has studied the Edge of Stability (EoS) phenomenon and it appears that training hyperparameters are typically selected in such a way that neural networks are often trained in the EoS regime. Nobody knows exactly how typical overparametrized neural networks are able to generalize as well as they do, but implicit regularization is theorized to play a role, and our work suggests that the implicit regularization may be enhanced by training in the EoS regime, which might partially explain why people train models in that way.
>
>
> *Regarding Sections 5-7:* We believe that the main point of confusion in the presentation, as raised by all the reviewer, comes from the fact that section 6 had to be subsection 5.2. Unfortunately, we accidentally made a subsection into a section. This has been solved right now and the section has been clarified further. Moreover, we updated the lemmas to slightly stronger versions proved in the newly designed Appendix C (since Reviewer 4 rightfully complained about it).
> Contentwise, we made clear that Section 5 deals with how the regimes of optimization change given the size of the learning rate and Section 6 regards a notion we introduced to prove linear convergence.
>
>
> We have corrected typographical and syntax errors throughout the manuscript and we believe we substantially improved readability and precision. Please see the new PDF uploaded.

---

> > ### Comment · Reviewer_ZTfs · 2024-11-22
> >
> > Thanks for the answer. I understood from your answer that there is a tradeoff between the simplicity of the model and the details of anylisis one can achieve.
> >
> >  I still think that calling _network_ a scalar function can be misleading, but overall I am more convinced that this paper could be a good contribution. Therefore, I raise my score to 6.

---

> > > ### Author Response · Authors · 2024-11-27
> > >
> > > Thanks a lot for the raise in score.
> > > Regarding "I still think that calling network a scalar function can be misleading": While we understand it is that people use networks mainly for high dimensional problems in practice, we want to point out that this is not misleading from the point of view of the optimization landscape, which is strongly influenced by the multiplicative/compositional structure of the model.
> > > Once again, we thank you for your insightful comments. We look forward to your further guidance to finalize our manuscript for publication in case you have more comments.

---

### Official Review · Reviewer_VT2a · 2024-11-04

**Soundness:** 3
**Presentation:** 1
**Contribution:** 2
**Rating:** 3
**Confidence:** 3

**Summary:**

This paper analyzes the effect of taking a large learning rate in training two-layer linear neural networks with single input and single output, formalized by $\mathbf{a}^\top\mathbf{b}$. First, the authors evaluate the imbalance between $\mathbf{a}$ and $\mathbf{b}$ and ensure that by taking a larger learning rate, GD converges to a more imbalanced solution.
The authors also analyze the convergence rate and show that a larger learning rate leads to a slower convergence speed.

**Strengths:**

The analysis of the EoS phenomenon is one of the significant topics in the deep learning theory literature. The convergence results exhibited in Theorem 1 and 2 provide informative insights into this problem.

**Weaknesses:**

While the topic treated in this paper is interesting, the latter section, specifically from section 5, needs to be completed. For example, I could not find the proof of Propositions 2 and 3 in the Appendix. Moreover, some results are demonstrated without detailed explanation. Regarding these drawbacks, I vote for rejection.

**Questions:**

The model treated in this paper is quite simple, while the same model has been treated in the existing literature. Is there any implication for more complex models, such as matrix factorization with multi-dimensional input?

---

> ### Author Response · Authors · 2024-11-21
> **Response to VT2a**
>
> We thanks a lot Reviewer VT2a for their comments.
>
> *Regarding the weaknesses:*
> - *Sections 5-7:* We believe that the main point of confusion in the presentation, as raised by all the reviewer, comes from the fact that section 6 had to be subsection 5.2. Unfortunately, we accidentally made a subsection into a section. This has been solved right now and the section has been clarified further. Moreover, we updated the lemmas to slightly stronger versions proved in the newly designed Appendix C (since Reviewer 4 rightfully complained about it).
>     Contentwise, we made clear that Section 5 deals with how the regimes of optimization change given the size of the learning rate and Section 6 regards a notion we introduced to prove linear convergence.
> - *Proofs of Proposition 2 and 3:* You are completely right that the proof of Proposition 2 and 3 were not there explicitly. They are deducible from a number of results in the appendix. We added in (Appendix D) a subsection with formal proofs.
> - *Detailed explanations:* Thanks a lot for pointing this out, and sorry.  We clarified our proofs adding much more details (as can be seen on the newly uploaded pdf). Thanks for raising this and sorry if this made you spend more time than necessary at reviewing our paper.
>
> We have corrected typographical and syntax errors throughout the manuscript and we believe we substantially improved readability and precision. Please see the new PDF uploaded.
>
>
> *Regarding your question:*
>
> It is true that the model we consider is quite simple, and has already been studied in various forms in the existing literature. That said, we believe that we have several significant contributions to make:
> - our analysis is more detailed than that of previous work (see discussion in section 1),
> - this additional detail allows us to prove a guarantee about the rate of convergence which is lacking in related work,
> - the detail also allows us to closely characterize the degree of implicit regularization caused by gradient descent, and
> - putting these together we are able to highlight an interesting phenomenon where convergence speed and implicit regularization apparently trade off against each other.
>
> While other papers may study the same model (or more complex versions of it), none of them include analysis at this level of detail, and therefore they do not arrive at our conclusions, which we think are interesting and could form the basis for additional study. Certainly, we would like to extend our analysis to more complex models, but this presents a significant technical challenge that must be left for future work.
>
>
>
>
> Thanking you again for your time we remain at your disposal for more comments and more suggestions on how to enhance our paper.

---

> > ### Comment · Reviewer_VT2a · 2024-11-25
> >
> > Thank you for the answer and for addressing my concerns about the writing after section 5.
> > I looked through the revised paper, including section D. The writing of Proposition 2,3 still has room for improvement. For example, at the end of section D.1, the authors state that “this lemma and the observation of what happens in the case of $Q = 0$ in Section E.2 prove Proposition 2”, but this statement seems vague. I could not find the point where you mean by “in the case of $Q=0$ in section E.2”.
> >
> > Moreover, I have several questions on Theorem 2. What do you mean by “it could be logarithmic slow”? It would be more helpful for readers if the authors provided a formal definition of this statement. Moreover, while the authors focus on the value $\min_t \lambda$ and construct the convergence analysis, analyzing $\sum_t \lambda$ would be more natural by the relationship (10). The relationship between the left and right figures in Figure 3 does not fully reflect the statement (even in the small step size case, it seems that the chaotic pattern appears in the right one). Does not this mean that the analysis focusing on $Q$ can not fully explain the convergence? I would appreciate it if the authors could address this point.

---

> > > ### Author Response · Authors · 2024-11-27
> > > **Response to Official Comment by Reviewer VT2a**
> > >
> > > Thank you for your continued feedback and for the additional scrutiny of section D. We appreciate a lot your guidance in enhancing our manuscript's clarity and rigor.
> > >
> > > *Clarification of Section References:*
> > > We apologize for the confusion regarding the section references in our previous draft.The reference should have been to section D.2, not E.2. We have updated the PDF. Please, review the modified document if you have time.
> > >
> > > *Theoretical Clarifications in Theorem 2:*
> > > Regarding ``it could be logarithmically slow,'', we understand that is not sufficiently defined, we will add in the appendix a definition of it. It means the sequence converges sublinearly and the limit of the rate of the updates is 1.
> > >
> > > *Choice of Analytical Focus:*
> > > Although may seem more natural, analyzing $\sum_t \lambda_t$ instead of $\min_t \lambda$ does not bring any real convergence speed benefit (only in terms of constants). Indeed the analysis would lead to the same kind of speed of convergence (linear).
> > > However, dealing with $\min_t \lambda_t$ facilitates deriving specific insights into the location of convergence, as detailed in Theorem 1. We believed was thus more powerful, although maybe less natural, to work with that quantity.
> > >
> > > *Regarding Figure 3*
> > > We do not believe that our analysis does not reflect Figure 3.
> > > - $\eta \lambda\leq 2$: The first part of Theorem 2 implies that the rate of convergence is decreasing for $1<\eta \cdot \lambda <2$ and increasing and for $0<\eta \cdot \lambda <1$. This is exactly what the picture shows, we thus believe Figure 3 here fully reflects the statement.
> > > Regarding Q: (9) and Theorem 1 suggest that $Q$ decreases as $\exp( - \eta^2 \sum_k \epsilon^2)$, for small $\eta$, the residuals convergence exponentially in time $1/\eta$. Thus $Q(\infty)/Q(0)$ is of size $ exp(-\eta \epsilon(0)^2) \sim (1 - 4\eta) \sim 1$.
> > > - $\eta \lambda> 2$: Chaotic behaviors are present, in particular in the first steps the dynamics get shot in other areas of the space: Here a worst-case-scenario convergence rate theorem will tell you that convergence could be slow, and that's exactly what Theorem 2 says. However, the initial increase in the residuals $\varepsilon$ due to this overshooting is greatly beneficial for $Q$, and this is what we can see in the LHS.
> > >
> > >
> > > Once again, we thank you for your insightful comments and hope that these revisions address your concerns effectively. We look forward to your further guidance to finalize our manuscript for publication.

---

> > > > ### Author Response · Authors · 2024-12-02
> > > >
> > > > Dear Reviewer VT2a,
> > > >
> > > > With the discussion period coming to an end, we were wondering if our comments and edit would be enough for you to raise your score.
> > > >
> > > > Thanks so much in any case,
> > > > Best,
> > > > The Authors

---

> > > > > ### Comment · Reviewer_VT2a · 2024-12-03
> > > > >
> > > > > Thank you for the answer and revision of the paper. Regarding comments from other reviewers, I believe this work could be immensely improved by further revision and discussion. Based on this, I would like to keep my score.

---

### Official Review · Reviewer_yA1T · 2024-11-06

**Soundness:** 2
**Presentation:** 1
**Contribution:** 2
**Rating:** 5
**Confidence:** 4

**Summary:**

The paper builds on recent studies of gradient descent on a two-layer linear network with scalar inputs and outputs.  The main results are on the location and speed of convergence of gradient descent.  By a careful analysis of the training dynamics which includes the edge of stability phenomenon, the main results elucidate the interplays among the learning rate, the implicit regularization, and the speed of convergence.  The theoretical results are supplemented by plots provided by numerical experiments.

**Strengths:**

This is a simple yet challenging setting, chosen well for this in-depth theoretical analysis.

The results are non-trivial and interesting, and proofs are provided in the appendix.

The plots from the experiments are detailed and complement the theoretical results well.

**Weaknesses:**

The grammar and spelling are poor throughout the paper, and especially in Sections 5-7, sometimes causing near unreadability.

In Sections 5-7, the presentation is confusing.  It is not even clear what the purpose of those sections is, I suppose they are meant to show some components of the proofs of Theorems 1 and 2, since they consist of only Definitions, Lemmas and Propositions, however that is not explained.  If that is true, then it is not clear how and to what extent Sections 5-7 provide outlines of the proofs of Theorems 1 and 2.

There are no suggestions for future work.

There are no details about the experiments, and no code as supplementary materials.

More minor issues:
- In the last paragraph on page 2, I think the multiplier of Q(t) in parentheses should be in absolute value.
- In equation (8), I am not sure where the $\sqrt{3}$ above $\approx$ comes from, and also what the meaning of writing a condition like this above $\approx$ is.
- In Theorem 2, after "If", what is $\epsilon$, is it meant to be $\epsilon(0)$?
- It seems Section 6 should have been Section 5.2.
- "Fig 3" versus "Figure 2", please be consistent.
- The citation of Wang et al. at the end of Section 6 should not be in parentheses.
- Proposition 3 contains $\eta < \eta$.
- Only one item in the References has a clickable link.

**Questions:**

In the second-level bullet points before Proposition 2, there are various claims about convegence, why are they true?

How is Proposition 3 "the formal and clean version of Theorem 2", e.g. it contains nothing like the last part of Theorem 2?  And in what sense is Theorem 2 informal and dirty?

The remarks after Proposition 3 seem to contradict the last part of Theorem 2, i.e. "we can prove that no matter the step size and the initialization we have linear convergence" is at odds with "we have convergence but it could be logarithmically slow"?

---

> ### Author Response · Authors · 2024-11-21
> **Response to yA1T**
>
> First of all we want to thank Reviewer yA1T a lot for the appreciation and the comments. We are very sorry if the fact that most of the weaknesses were about presentation and text slowed down the review process. We put an effort on addressing the concerns raised:
>
> *Regarding text and presentation:*
> - We have corrected typographical and syntax errors throughout the manuscript to improve readability and precision, thanks again for pointing out. Please see the new PDF uploaded for these changes.
> - We believe that the main point of confusion in the presentation, as raised by all the reviewer, comes from the fact that section 6 had to be subsection 5.2. Unfortunately, we accidentally made what should have been a subsection into a section. This has been solved right now and the section has been clarified further. Moreover, we updated the lemmas to slightly stronger versions proved in the newly designed Appendix C (since Reviewer 4 rightfully complained about it).
>     Contentwise, we made clear that Section 5 deals with how the regimes of optimization change given the size of the learning rate and Section 6 regards a notion we introduced to prove linear convergence.
> - Minors: Details on the experiments and the code appeared the supplementary material. Suggestions for future work appeared in the conclusions. Most references have now clickable links and we will add it to the others.
> - Thanks for pointing out all the typos and inconsistencies, we should have fixed and clarified everything.
>
>
>
> *Technical Questions:*
> - *Claims about convergence:* We discussed them and proved them in Appendix D. Essentially, in those cases the dynamics is trapped in a one dimensional manifold that leads somewhere specific (as in a saddle). We are claiming that the speed of convergence is linear within that constrained manifold.
> - *Proposition 3 vs Theorem 2, 1:* You are completely right, we removed this sentence. Proposition 3 is more pleasure
> - *Proposition 3 vs Theorem 2, 2:* We bounded differently the learning rate in the two results. Precisely, Proposition 3 only applies to smaller learning rates than Theorem 2.
>
> We hope we answered all the questions.
> While thanking again for your review and comments, we remain at disposal for further suggestions to enhance our manuscript and improve its quality and impact.

---

> > ### Comment · Reviewer_yA1T · 2024-11-21
> >
> > Thank you for this response, and for revising the paper.  I shall see what the other reviewers think, and then consider whether to modify my score.

---

> > > ### Author Response · Authors · 2024-12-02
> > >
> > > Dear Reviewer yA1T,
> > >
> > > With the discussion period coming to an end, we were wondering if our comments and edit would be enough for you to raise your score.
> > >
> > > Thanks so much in any case,
> > > Best,
> > > The Authors

---

### Meta-Review · Area_Chair_CR11 · 2024-12-05

**Metareview:**

The study provides analytical insights into the solutions to which gradient descent converges when applied to two-layer linear networks with a single input and output, emphasizing scenarios involving large step sizes. While the reviewers found the topic and its connection to EoS interesting and timely, they raised concerns regarding the writing and clarity. In particular, some of the proofs were found unreadable, erroneous, or even missing, as acknowledged by the authors. Finally, some reviewers considered the model overly simplistic, questioning the broader relevance and generalizability of the results. The authors are encouraged to address the important feedback provided by the knowledgeable reviewers.

**Additional Comments On Reviewer Discussion:**

Strong objections to acceptance were raised on account of the issues outlined in the MR, which remained unresolved despite the discussions.

---

### Decision · Program_Chairs · 2025-01-22

Reject